# Controlling motor neurons of every muscle for fly proboscis reaching

**Claire E McKellar[1,2]\*, Igor Siwanowicz[1], Barry J Dickson[1,3], Julie H Simpson[1,4]‡\***

[1]Janelia Research Campus, Howard Hughes Medical Institute, Ashburn, United States; [2]Princeton Neuroscience Institute, Princeton University, Princeton, United States; [3]Queensland Brain Institute, University of Queensland, St Lucia, Australia; [4]Dept. of Molecular Cellular and Developmental Biology, University of California Santa Barbara, Santa Barbara, United States

**Abstract** We describe the anatomy of all the primary motor neurons in the fly proboscis and characterize their contributions to its diverse reaching movements. Pairing this behavior with the wealth of *Drosophila's* genetic tools offers the possibility to study motor control at single-neuron resolution, and soon throughout entire circuits. As an entry to these circuits, we provide detailed anatomy of proboscis motor neurons, muscles, and joints. We create a collection of fly strains to individually manipulate every proboscis muscle through control of its motor neurons, the first such collection for an appendage. We generate a model of the action of each proboscis joint, and find that only a small number of motor neurons are needed to produce proboscis reaching. Comprehensive control of each motor element in this numerically simple system paves the way for future study of both reflexive and flexible movements of this appendage.

**\*For correspondence:**
claireem@princeton.edu (CEMK);
jhsimpson@ucsb.edu (JHS)

**Present address:** †Princeton Neuroscience Institute, Princeton University, Princeton, United States; ‡University of California Santa Barbara, Molecular Cell and Developmental Biology, Santa Barbara, United States

**Competing interests:** The authors declare that no competing interests exist.

## Introduction

To interact with the world, many organisms rely on an ability to reach an appendage towards a target, yet much remains to be discovered about how neural circuits control this prototypical goal-directed behavior. Here we investigate the fly proboscis (mouthparts), a segmented appendage homologous to a limb (*Bridges and Dobzhansky, 1933*). The proboscis is capable of directed reaching towards a specifically placed food target, using gustatory information from taste receptors on the legs as they move and contact food (*Dethier, 1976*). Besides feeding, the proboscis also extends during courtship (*Sturtevant, 1915*), grooming (*Szebenyi, 1969*), flight-associated respiration (*Lehmann and Heymann, 2005*), and wing expansion after eclosion (*Baker and Truman, 2002*), suggesting that it is a multifunctional appendage that can be employed flexibly.

A good starting point for mapping circuits underlying motor control is to identify motor neurons, since they are the outputs upon which higher order circuits must act. In adult flies, most muscles are innervated by only one or a few motor neurons (see, for example, *Baek and Mann, 2009*; *O'Sullivan et al., 2018*; *Rajashekhar and Singh, 1994*; *Trimarchi and Schneiderman, 1994*). Identifying many or all of the motor neurons involved in proboscis extension should be feasible with available genetic tools. Groups of neurons can be activated and silenced to determine their behavioral phenotypes using collections of GAL4 lines (*Brand and Perrimon, 1993*), which reproducibly target different groups of neurons. Previous efforts using GAL4 lines with broader expression patterns, or stochastic techniques to refine GAL4 lines, identified some of the critical motor neurons (*Gordon and Scott, 2009*; *Schwarz et al., 2017*). Here we use the split GAL4 technique, which intersects two expression patterns to genetically target only the cells in common between the two (*Luan et al., 2006*), to provide more precise and reproducible targeting of particular neurons. We create a collection of split GAL4 fly strains to identify and separately target the primary motor neurons for every proboscis muscle.

To support this collection of genetic reagents, we provide detailed anatomy of the proboscis from a variety of imaging techniques. We show each proboscis muscle in new detail, propose a systematic nomenclature to resolve earlier discrepancies, and use muscle insertion sites to predict function. We predict that 8 of the 16 proboscis muscles should be involved in proboscis positioning (such as reaching), and the other half should be involved in the proboscis' other main function, pumping food through the pharynx (investigated by others; *Flood et al., 2013*; *Manzo et al., 2012*; *Tissot et al., 1998*). Here we focus on the predicted positioning muscles and define their roles by genetic activation and silencing of their motor neurons. Altogether, we achieve genetic control over every muscle involved in this form of directed reaching, and provide detailed anatomy of this appendage.

## Results

### Anatomy of the proboscis and its joints

Our first goal was to use updated imaging techniques to display the overall anatomy of the proboscis (*Figure 1*).

Work that predated confocal microscopy (*Miller, 1950*) showed that the proboscis is extended during feeding. Food is sucked in through the lip-like labella at the tip, then pumped through the pharynx, traveling through two segments of the proboscis, first the haustellum then the rostrum. Inside the head, food then enters the esophagus, which passes through a foramen in the brain, and carries food towards the gut. *Figure 1A* shows these structures by confocal imaging of thick sections of fly heads with the proboscis in different degrees of extension.

In the retracted state, the rostrum rests near the brain inside the head, and the haustellum is folded up against the rostrum (*Figure 1A*). When the rostrum extends, it is protracted ventrally out of the head capsule. The outer cuticle of the rostrum is largely unsclerotized, giving it flexibility to fold like skin; support for several muscles therefore comes from an internal skeleton of rigid cuticular structures (*Figure 1A–C*). The haustellum extends and flexes from its joint with the rostrum. These movements were confirmed in live flies by filming proboscis extension in a synchrotron particle accelerator, using x-rays to render flies partly transparent and show the movement of the segments inside the head capsule (*Video 1*).

To locate proboscis joints with standard light cameras, proboscis extension was triggered in tethered flies by stimulating leg gustatory receptors with a droplet of sucrose (approximate joint locations marked, *Video 2*). Gluing the head and one proboscis joint shows only the movement of the remaining joint: the rostrum pivots around a point on the anterior head that we designate the rostrum joint (*Video 3*, and also visible in *Video 1*), and the haustellum pivots around a point near the rostrum-haustellum junction (*Video 4*). We digested soft tissues from the thick slices of fly heads, leaving the rigid external and internal cuticle to visualize the locations of the articulation points (*Figure 1B,C*). Without implying genetic homology, a useful analogy is that the rostrum and haustellum move somewhat like the human upper arm and lower arm, the first segment swinging away from the body and the second unfolding from the first. (The movements of the rostrum are called protraction vs. retraction, whereas the movements of the haustellum are extension vs. flexion [*Miller, 1950*]). For this study, we focused on movements in only two dimensions, in the vertical plane - reaching up and down with respect to the body axis.

### A quantitative description of proboscis reaching

Armed with this demonstration of how the segments of the proboscis subserve proboscis extension, we sought quantitative proof that the *Drosophila* proboscis is capable of directed reaching. We had observed that, for a freely moving fly to feed, the proboscis must extend ventrally if the food is on the surface on which the fly is standing (*Figure 2A*, *Video 5*). In contrast, we find that a male courting a female on a flat surface typically extends his proboscis anteriorly towards the female for courtship licking (*Figure 2B*, *Video 6*). To quantify reaching in these particular situations, the angles of the head (*Figure 2C*), rostrum (*Figure 2C*), and haustellum (*Figure 2D*) were measured. Both the head and the rostrum were angled higher (more anteriorly) in the courtship experiments than in feeding, on average (*Figure 2E–F*). The haustellum was more extended for ventral feeding, and more flexed for anterior courtship licking (*Figure 2G*). The proboscis is thus capable of reaching to

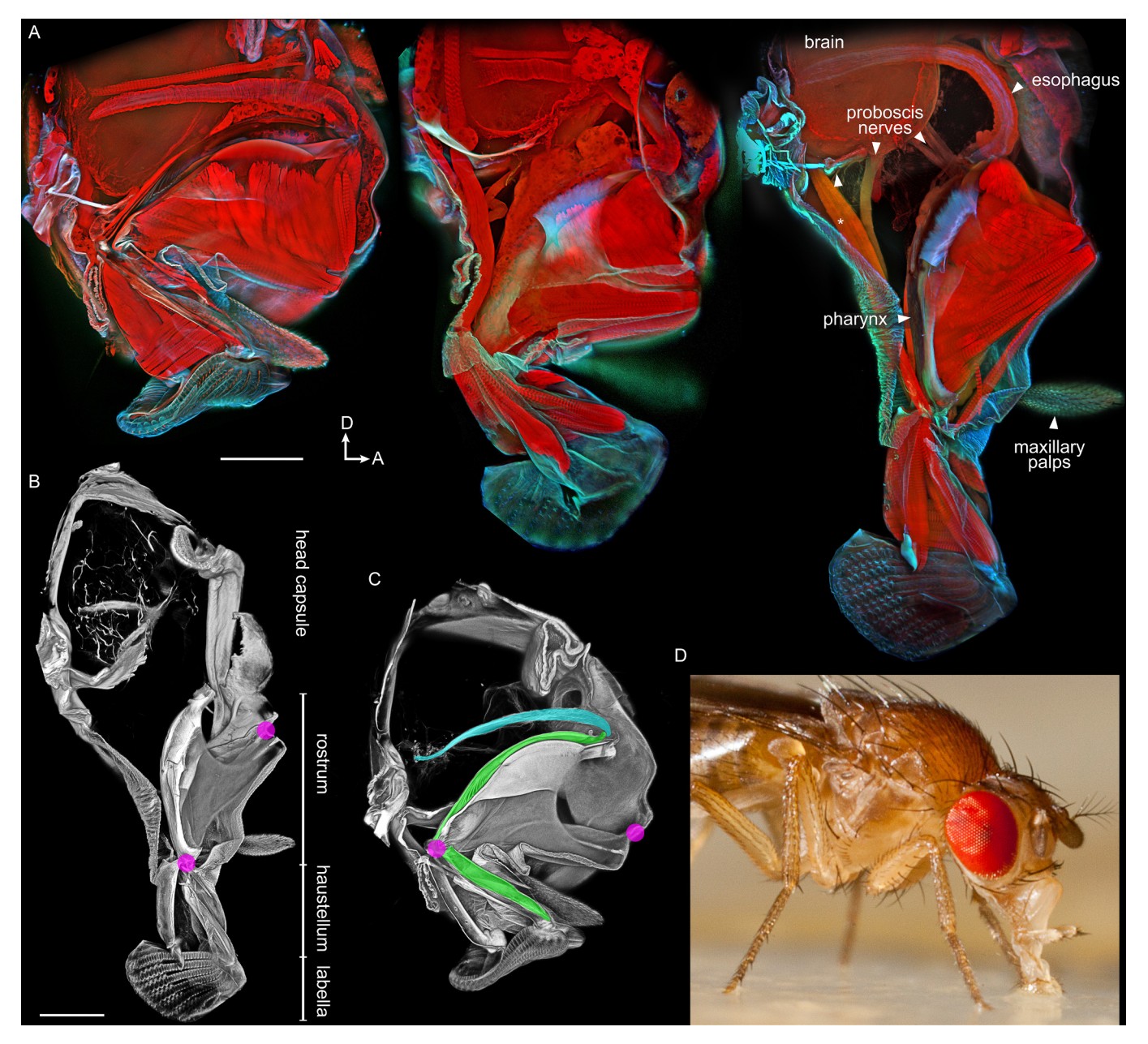

**Figure 1.** Anatomy of the proboscis. (**A–C**) Thick sagittal sections of fly heads with proboscis in various states of extension. Scale bars: 100 µm. Arrows, D: dorsal, A: anterior. (**A**) Muscles and other internal tissues stained with phalloidin (red). Cuticle and sclerites stained with calcofluor white (cyan). (**B–C**) Soft tissues digested away to reveal rigid external and internal cuticle. Magenta: locations of proboscis joints. Green: pharynx. Cyan: esophagus. (**D**) Lateral view of a feeding fly, showing proboscis touching surface near forelegs.

ethologically relevant targets in different locations, making use of the head, the rostrum and the haustellum.

This differential reaching may not be inherently due to feeding or courtship, but to the location of the targets; perhaps the food was just located more ventrally than the female, prompting more ventral proboscis extension. We thus quantified proboscis reaching to food targets placed in different locations with respect to the head of a tethered fly. When the sucrose was placed low or high (more ventral or more anterior, *Figure 2H–J*), the proboscis was indeed capable of low vs. high reaching (*Figure 2K*, *Video 7*), with decent aim (*Figure 2L*). Although individual flies could angle joints

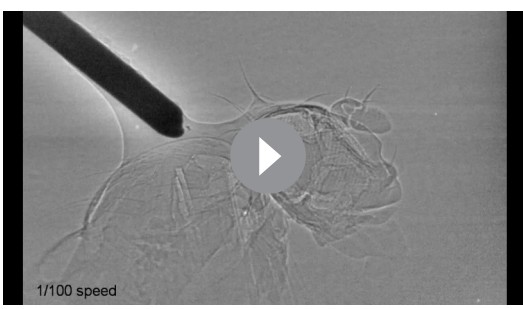

**Video 1.** Synchrotron x-ray video of proboscis extension. Looping video of proboscis extension in a fly tethered with head glued, in a synchrotron x-ray beam, sagittal view, 1/100 speed. This example primarily shows movement of the rostrum (labeled magenta in the later portion of the video), pivoting around the joint marked by crosshairs, with little extension of the haustellum and labella (green).
https://elifesciences.org/articles/54978#video1

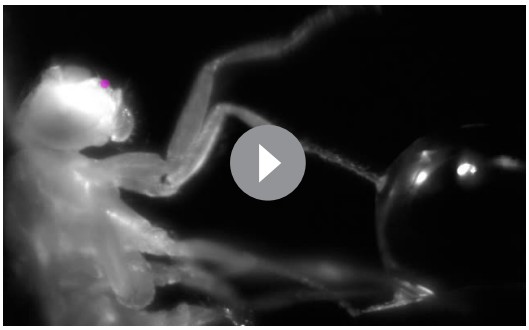

**Video 2.** Proboscis joint movements in response to sucrose. Sagittal view of a tethered fly (anterior: up) reaching the proboscis towards a droplet of sucrose presented to the legs. 1/30 speed. Magenta dots: approximate locations of two proboscis joints.
https://elifesciences.org/articles/54978#video2

differentially to different targets (gray lines, *Figure 2M–O*), only the haustellum showed a mean difference across target positions (black bars, *Figure 2M–O*), suggesting that the haustellum can be independently controlled to aid reaching. Each of the three body parts (rostrum, haustellum and head) is capable of at least some degree of independent movement, since if two of the three parts were glued, the remaining part could still move (*Videos 3*, *4* and *8*). Reaching was guided not by a visual target (due to filming under invisible infrared light) but by contact between the waving legs and the sucrose target; the relevant sensory stimuli are therefore: 1) taste, and 2) proprioception about leg position at the time of sucrose contact.

## An updated inventory of proboscis muscles

To explore how anatomy supports this reaching, we adapted a clearing technique (*Ott, 2008*) to stain and image whole-mount heads, and reveal intact muscles and their motor neurons with high clarity (*Video 9*). Using 3D segmentation, we traced every muscle in the proboscis of the mature adult (*Figure 3A*, *Video 10*). We find a total

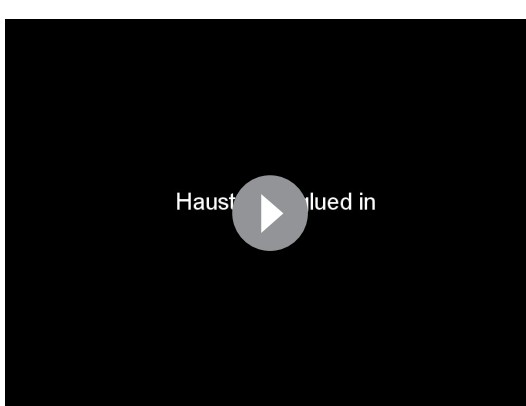

**Video 3.** Rostrum joint movement. Proboscis movement restricted to rostrum joint by gluing haustellum (in or out) and head. Sagittal view of tethered fly, anterior: up, 1/3 speed. Magenta dot: approximate location of rostrum joint.
https://elifesciences.org/articles/54978#video3

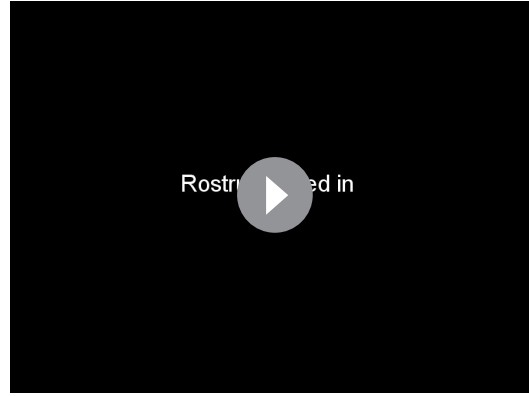

**Video 4.** Haustellum joint movement. Proboscis movement restricted mainly to haustellum joint by gluing rostrum (in or out) and head. (Rostrum glued along one surface, not completely immobilized). Sagittal view of tethered fly, anterior: up, 1/3 speed. Magenta dot: approximate location of haustellum joint.
https://elifesciences.org/articles/54978#video4

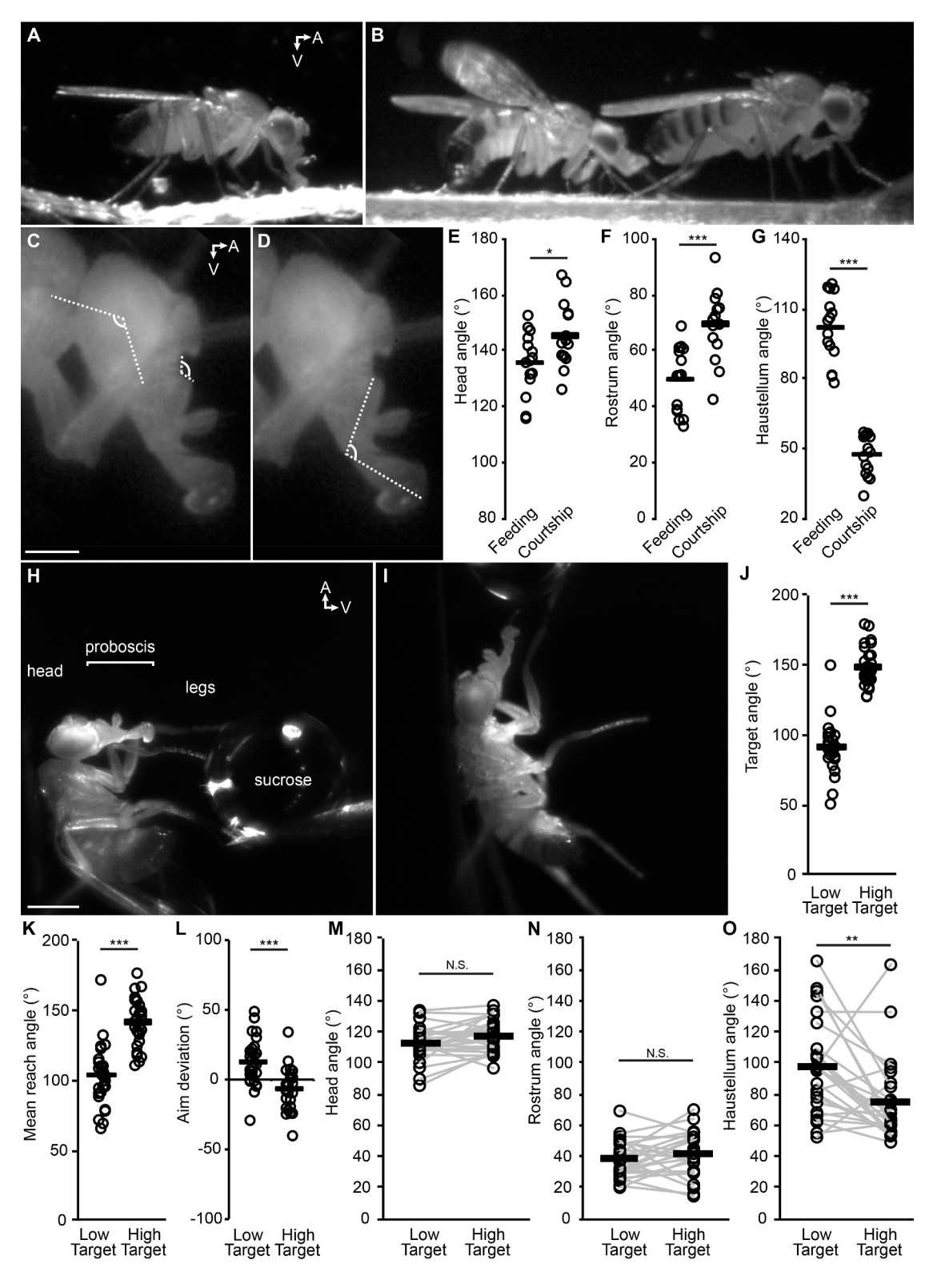

**Figure 2.** The fly proboscis as a model system for directed reaching. (**A**) A fly extending the proboscis towards food on the surface of an experimental chamber (sagittal view). (**B**) A male fly (left) courting a female (right), extending his proboscis towards the female's posterior. (**C**) Points used for measurement of angles of the head (connected by the longer lines) and rostrum (shorter lines). Scale bar: 200 μm. (**D**) Measurement of haustellum angle. (**E–G**) Angle of head (**E**), rostrum (**F**) and haustellum (**G**) in males feeding or courting on a flat surface. n = 15 flies per condition. T-test (unpaired)

*Figure 2 continued on next page*

*Figure 2 continued*

*p<0.05, **p<0.01, ***p<0.001, N.S.: not significant. (H–I) Proboscis extension in response to a low (H) or high (I) sucrose droplet presented to the legs of a tethered fly. Scale bar: 500 µm. (J–L) n = 28 males each presented with sucrose once in low position, once in high. T-test (paired). (J) Angle of target from the fly at frame of first leg contact, when target placed in low or high positions, where 90° would be directly ventral to the eye. (K) Mean reach angle: angle from the posterior-anterior axis of the fly to the proboscis tip, averaged over proboscis extension bout. (L) Aim deviation: reach angle minus target angle. (M–O) Joint angles scored 200 ms after beginning of PE: head (M), rostrum (N), haustellum (O).

count of 16 proboscis muscles. (Other muscles are present at eclosion, possibly to help the fly push out of the puparium, but subsequently degenerate, *Miller, 1950*).

Different names are given to identical muscles in recent papers (*Flood et al., 2013*; *Schwarz et al., 2017*), therefore we propose a nomenclature with new suffixes to resolve confusion, consisting of informative letters instead of numbers (*Figure 3B*, *Table 1*). For example, the former 2–1 and 2–2 are now 2D and 2V, referring to more dorsal or ventral insertion sites. A prefix of 'm' refers to muscle, 'mn' refers to motor neuron(s) for that muscle. (We show below separable neuronal control of the newly found m3L and its neighbor m3M; however their adjacent insertions and origins lead us to define these as separate groups of fibers in a single muscle, m3. Aside from this exception, suffixes denote separate muscles).

## Achieving genetic control of proboscis motor neurons

To understand the neural control of these muscles, we created genetic reagents to target every primary proboscis motor neuron type. By inspecting thousands of Rubin and VT-GAL4 lines (*Jenett et al., 2012*; *Pfeiffer et al., 2010*; *Tirian and Dickson, 2017*), we selected 100 lines with neurons resembling candidate proboscis motor neurons (characterized by large somata in the SEZ and axons in the SEZ nerves), to stain with the new head clearing technique. Of these, 77 indeed had motor neurons projecting to proboscis muscles (*Table 2*; see example in *Video 9*). Few contained only a single motor neuron type, and all contained many other cells in the brain.

To obtain more specific expression patterns, we intersected these lines using the split GAL4 genetic technique (*Luan et al., 2006*). A many-to-many approach was taken, staining several combinations of lines that each contained a motor neuron of interest (along with other cells). A final collection of 17 split GAL4 lines was selected, most of which target a single motor neuron type (with exceptions being one line targeting both mn11V and 12D, and three lines for mn3) (*Table 1*). Proboscis staining showed each line's muscle targets (*Figure 4*), with good reproducibility of expression patterns across specimens (*Table 1*). Central nervous system (CNS) staining showed that besides expression in proboscis motor neurons, these 17 were otherwise quite sparse (*Figure 4—figure supplement 1*), having been selected from a total of 389 combinations by CNS staining.

Our goal in building this collection of lines was to target every primary excitatory motor neuron type of the proboscis. In the larva, the primary excitatory motor neurons are called Type Ib and Is; the motor neurons here most closely resemble the majority of larval Type Ib neurons in that they innervate only a single muscle, unlike the more promiscuous larval type Is (*Budnik and Ruiz-*

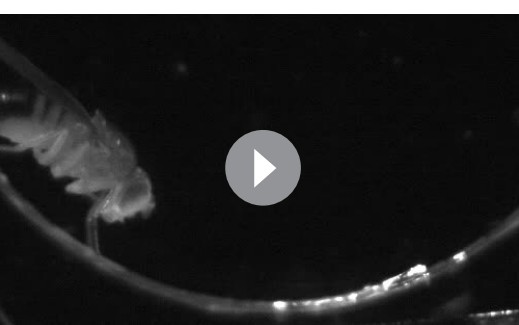

**Video 5.** Proboscis extension during feeding. A wildtype male feeding from sucrose painted onto wall of chamber (bright region) extends the proboscis ventrally towards the food. 1/1 speed.
https://elifesciences.org/articles/54978#video5

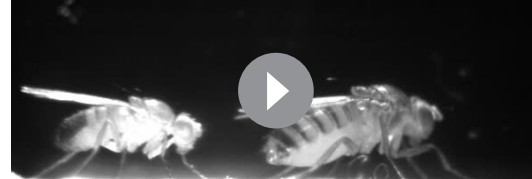

**Video 6.** Proboscis extension during courtship. A male (left) courting a female (right) on a flat surface extends the proboscis anteriorly towards the female. 1/10 speed, wildtype flies.
https://elifesciences.org/articles/54978#video6

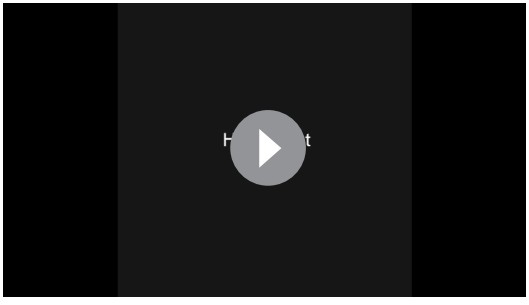

**Video 7.** Proboscis reaching to targets in different locations. Proboscis and head movements reaching towards a sucrose target in high or low (anterior or ventral) locations. Sagittal view of tethered fly, anterior: up, 1/1 speed.
https://elifesciences.org/articles/54978#video7

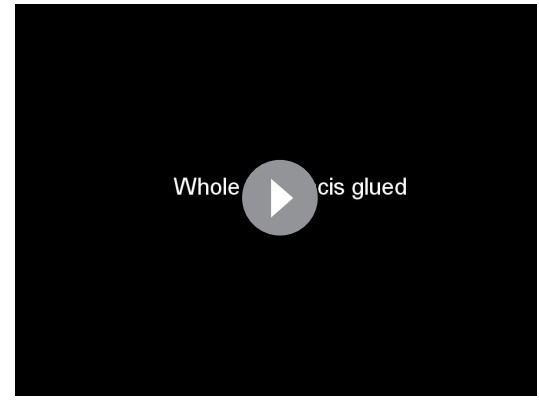

**Video 8.** Head movement. Head movement towards sucrose when both rostrum and haustellum joints are glued (not labella). Sagittal view of tethered fly, anterior: up, 1/3 speed.
https://elifesciences.org/articles/54978#video8

*Canada, 2006*). (We did not seek modulatory motor neurons, analogous to type II and III in larva, which are characterized by small boutons on thin axons projecting to multiple muscles [*Prokop, 2006*], and are known to exist in the adult proboscis [*Pauls et al., 2018*].) To determine whether this collection was complete, we stained each line with nc82, considered a pan-synaptic marker (*Wagh et al., 2006*). All the boutons on the relevant muscle were indeed completely occupied by the labeled motor neurons (examples in *Figure 4—figure supplement 2, N* in *Table 1*), with two exceptions: muscle 3 is covered by separate lines (mn3M and mn3L) so was not expected to show complete occupancy, and mn10 was never labeled bilaterally in the split GAL4 line we selected (see Methods). All other synaptic sites were occupied, and since the synaptic marker does label primary motor neuron synapses, we conclude that this collection includes every primary motor neuron. The coverage of all a muscle's synaptic sites by only one or a few

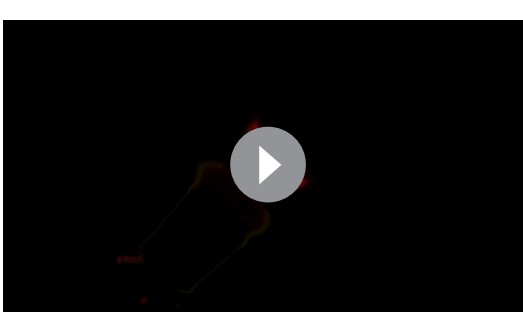

**Video 9.** Confocal stack showing an example of raw data from the head clearing technique, first in composite then each channel individually. Red: muscles stained with phalloidin. Green: GFP stain of several motor neuron types genetically targeted in this example, projecting to the rostrum. Blue: synapses stained with nc82.
https://elifesciences.org/articles/54978#video9

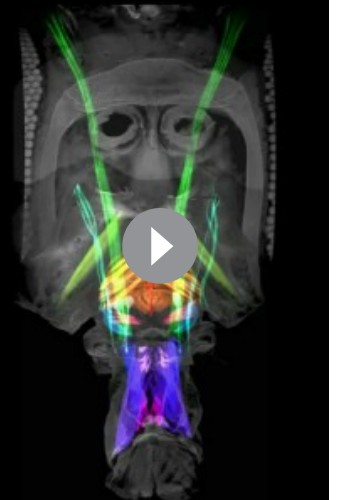

**Video 10.** A fly head imaged by cuticle autofluorescence (gray) with proboscis pointing downwards and eyes cropped out of image at sides. All muscles were stained with phalloidin (stain not shown), then traced in segmentation software to display every proboscis muscle in a different color, first overlaid, then individually. Last: approximate path of pharynx. (Pharynx is a more complex shape than shown here but is only visible as negative space with this staining, difficult to trace). The brain can be seen in dark gray within the head, with hook-shaped sclerites underneath it. Fat body and air sacs not shown.
https://elifesciences.org/articles/54978#video10

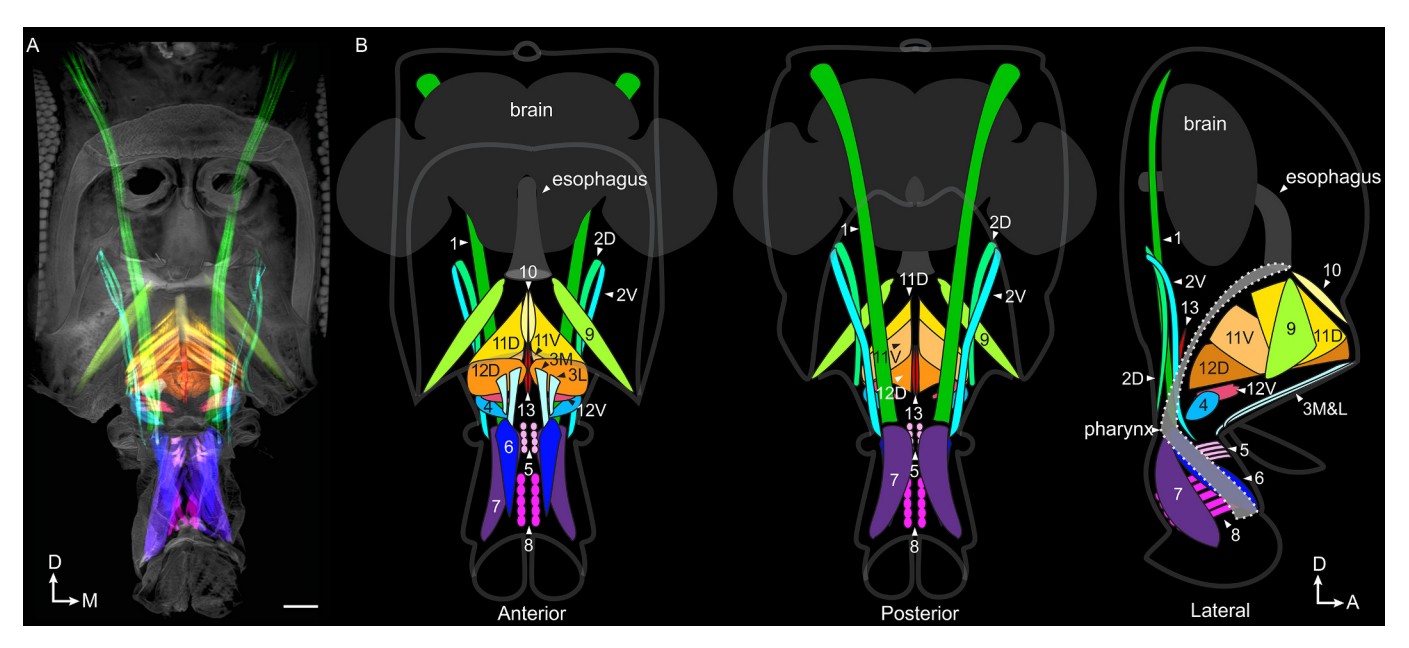

**Figure 3.** Proboscis muscles. (**A**) Frontal view of head with traced proboscis muscles, from clearing technique and segmentation software. Eyes removed at sides, and antennae removed, leaving two holes seen in upper center. Scale bar: 50 µm. (**B**) Schematics from different views as noted, showing proboscis muscles, brain, esophagus and pharynx. Pharynx superimposed for visibility (approximate outline: dotted line). D: dorsal, M: medial, A: anterior.

motor neurons per hemisphere shows that single motor neurons must branch to many or all of the separate fibers that make up a muscle (*Figure 4—figure supplements 1* and *2*).

## Organization of motor neuron dendrites in the brain

While the axons of motor neurons innervate muscles, their dendrites are located in the subesophageal zone of the brain. The organization of these dendrites was examined by segmenting them from split GAL4 images (or from stochastic labeling of split GAL4s, in cases where neighboring cells in the brain made identities uncertain), to give a clear view of every defined motor neuron type (*Figure 5*, *Video 11*).

Each line has only one or a few bilateral motor neuron pairs (*Figure 5*) to cover its muscle. All dendritic arbors except salivary mn13 fall into one of two regions, dorsal or ventral, which partially overlap (*Figure 5B*). These divisions did not correlate with muscles found in the rostrum vs. haustellum (*Table 1*). Instead, with few exceptions, the dorsal region was occupied by motor neurons with pharyngeal insertion sites, whereas the ventral region was generally occupied by non-pharyngeal motor neurons (*Table 1*). Pharyngeal and non-pharyngeal motor neurons also tended to have dorsal and ventral somas, respectively, and axons leaving the brain through dorsal and ventral proboscis nerves ('pharyngeal' and 'compound labial' nerves, respectively, *Hartenstein et al., 2018*), with certain exceptions (*Figure 5B*, *Table 1*).

The segmented motor neuron dendrites permit computational registration with images of other feeding-related neuron classes, to determine proximity between them (although not synaptic connectivity). Sweet inputs avoid the ventral arbors of the rostrum protractor's mn9, passing through two circular gaps to terminate more dorsally (*Figure 5—figure supplement 1A,B*), consistent with their known lack of contact with mn9 (*Gordon and Scott, 2009*). Sweet inputs likewise pass through gaps in the arbor of mn1 (*Figure 5—figure supplement 1C,D*), and also do not overlap with a pharyngeal neuron mn11D (*Figure 5—figure supplement 1E,F*). They do lie somewhat near the salivary mn13 (*Figure 5—figure supplement 1G,H*).

The circular gap in motor neuron arbors is particularly evident when several motor neurons (mn1, 9, 11D and 13) are overlaid (*Figure 5—figure supplement 1I*). We have previously shown that eye

**Table 1.** Muscle names and the best split GAL4 lines that target their motor neurons.

Muscle names in different papers, plus our split GAL4s targeting each muscle, motor neurons present in those lines, and location of motor neuron dendrites, soma and which proboscis nerve the axon uses. In split GAL4 names, D = dorsal, V = ventral, M = medial, L = lateral. In almost every case, when the motor neurons were present, they occupied 100% of the NMJs on the relevant muscle, with the exceptions of 3 and 10, described in the text. Former 12–1 is more similar to 11 than to 12–2; renamed as 11V. 12D and 12V are named for proximity, not implying related function. Yellow: positioning muscles that do not insert on the pharynx. Green: pharyngeal muscles.

| Miller, 1950 | Rajashekhar and Singh, 1994 | Flood et al., 2013 | Schwarz et al., 2017 | Muscle location | Split GAL4 | AD split half | DBD split half | MNs in # proboscis sides | Dend-rites | Soma | Nerve |
|---|---|---|---|---|---|---|---|---|---|---|---|
| 1,lateral labial adductor m. | retractor of rostrum | | 1 | rostrum | mn1 | VT043075 | VT019731 | mn1 in 18/18 | vent | vent | vent |
| 2,maxillary retractor m. | not investigated | | 2–1 | rostrum | mn2D | GMR11F07 | VT064563 | mn2D in 16/16 | vent | vent | vent |
| not known | not known | | 2–2 | rostrum | mn2V | VT064563 | GMR13E04 | mn2V in 12/12 | vent | vent | vent |
| 3 | flexor of labrum | | 3 | rostrum | mn3M &1 | VT025784 | VT063630 | mn3M in 4/4 mn1 in 4/4 | vent | dors | ND |
| 3 | flexor of labrum | | 3 | rostrum | mn3M&7 | VT063630 | GMR75F02 | mn3M in 12/12 mn7 in 6/12 | vent | dors | ND |
| not known | flexor of labrum | | not known | rostrum | mn3L | VT031145 | GMR89F06 | mn3L in 10/10 | vent | dors | dors |
| 4,maxillary m. | not investigated | | 4 | rostrum | mn4 | GMR48H12 | GMR45G01 | mn4 in 10/10 | vent | vent | vent |
| 5,labral compressor m. | not investigated | | 5 | haustellum | mn5 | VT033616 | VT043145 | mn5 in 10/10 | vent | dors | dors |
| 6,palpal m. | retractor of paraphysis | | 6 | haustellum | mn6 | GMR18B07 | GMR81B12 | mn6 in 10/10 | vent | vent | vent |
| 7,palpal m. | retractor of furca | | 7 | haustellum | mn7 | VT014959 | VT001484 | mn7 in 9/10 | vent | vent | vent |
| 8,transverse labial m. | transverse m. of haust. | | 8 | haustellum | mn8 and 7 | VT027168 | VT015822 | mn8 in 15/16 mn7 in 7/16 | vent | vent | vent |
| 9,lateral pharyngeal m. | protractor of fulcrum | | 9 | rostrum | mn9 | VT061715 | VT005008 | mn9 in 12/12 | vent | dors | dors |
| 10,median pharyngeal m. | median pharyngeal m. | | 10 | rostrum | mn10 | GMR14H09 | VT020713 | mn10 in 6/12 | dors | dors | dors |
| 11, pharyngeal m. | dorsal pharyng. dilator | m.11 | 11–1,11-2 | rostrum | mn11D | VT020737 | GMR10B11 | mn11D in 10/10 | dors | dors | dors |
| 12,cibarial m. | ventral pharyng. dilator | m.12–1 | 12–1 | rostrum | mn11V and 12D | VT050240 | GMR10E04 | mn11V in 21/22 mn12D in 21/22 | dors | dors | dors |
| 12,cibarial m. | not known | m.12–2 | 11–3 | rostrum | mn11V and 12D | VT050240 | GMR10E04 | mn11V in 21/22 mn12D in 21/22 | dors | dors | dors |
| not known | not known | | 12–2 | rostrum | mn12V | GMR80D06 | GMR75F02 | mn12V in 14/14 | dors | dors | dors |
| 13,dorsal salivary m. | not investigated | | 13 | rostrum | mn13 | VT043700 | VT034258 | mn13 in 12/12 mn4 in 4/12 | dors | vent | vent |

mechanosensory bristle neurons project to the ventral SEZ (*Hampel et al., 2017*), and now find that the glomerular-like terminals of these mechanosensory inputs occupy the circular gaps in motor

**Table 2.** Head stain results from 100 GAL4 lines.

Lines from Rubin (GMR) and Dickson (VT) collections, in attP2 landing site, showing motor neurons found in proboscis. Note: this clearing technique showed more motor neuron types than were previously known in two lines, GMR18B07 and GMR81B12 (*Schwarz et al., 2017*).

| GAL4 line | Motor neurons in proboscis | GAL4 line | Motor neurons in proboscis | GAL4 line | Motor neurons in proboscis |
|---|---|---|---|---|---|
| GMR10B11 | 11 or 12D,12V | GMR32D10 | none | VT037554 | none |
| GMR10E04 | 2V,7,8,11,12 | GMR41G04 | 1 or 2?,6,7, 8, 9, 11, 12? | VT037583 | 1,6 |
| GMR10E06 | none | GMR70E08 | 9 | VT037859 | 1,4,7? |
| GMR11B04 | 1,2V,4,6,7,10,11,12,13 | GMR72D06 | 2V | VT038335 | none |
| GMR11C05 | 9,11D | GMR75F02 | 3M,7,12V | VT039475 | none |
| GMR11D03 | 11 or 12? (stochastic) | GMR78E09 | 5,9,3? | VT041887 | none |
| GMR11D04 | 2(D?),7,8 | GMR81B12 | 6,10,1 stochastic | VT042739 | none |
| GMR11D09 | none | GMR89F06 | 3,5 | VT043075 | 1 |
| GMR11F07 | 1,2D,4,9,11,12V | VT000831 | 6,7,8 | VT043088 | 1,7 |
| GMR11F12 | 6 | VT001484 | 3,7 | VT043145 | 5,9,12V |
| GMR11G01 | 4,8 | VT003229 | 1,4,11 | VT043147 | 2V |
| GMR12E03 | 2V | VT005008 | 9 | VT043700 | 2D,6,13 |
| GMR13B02 | 3L,4 | VT011965 | 1,6,7 | VT049356 | none |
| GMR13E04 | 2V, 8? | VT013506 | 11 | VT049481 | 1,4,8,10 |
| GMR13F07 | none | VT013605 | 1,2V,6,7,9,10, (12 or 3),13 | VT049727 | none |
| GMR14H09 | 10 | VT014959 | 6,7,8 | VT050217 | 11 |
| GMR15E02 | 1,2D,2V,3L,4,5,6,7,12V | VT015822 | 1,7,8 | VT050240 | 3,11,12 |
| GMR15E10 | none | VT017933 | 8 | VT050663 | 4,6,7 |
| GMR17D03 | 9,10 | VT019731 | 1,7 | VT055404 | none |
| GMR18A08 | none | VT020713 | 10 | VT056658 | none |
| GMR18B07 | 7(stochastic),8(stoch.),9,10(dim),11,12,12V | VT020737 | 11 | VT057137 | 2D,2V |
| GMR18D09 | none (in brains, one stochastic MN) | VT022244 | none | VT057237 | 6,7,8 |
| GMR18G02 | 1,2D,2V,4,5,6,7,8 | VT023789 | 1,7,8,9 | VT057379 | 6 |
| GMR19A06 | 11 | VT025784 | 1,3 | VT058488 | 1,4,7,13 |
| GMR19G04 | 1,2D,2V,3L,4,5,7,10,11,12V | VT026026 | 8 | VT059784 | 10,11,12 |
| GMR20E07 | 6 | VT027168 | 7,8 | VT061715 | 9 |
| GMR20E09 | 2D,2V | VT031145 | 2V,3 | VT062553 | 1,6,7,13 |
| GMR20G03 | none | VT031157 | 11,12V | VT063219 | 1,6,13 |
| GMR21A10 | none | VT031562 | 4,12V | VT063302 | none |
| GMR23C08 | 4 | VT032912 | 2D,2V,4,8 | VT063630 | 1,2V(stochastic),3,4,7,8 |
| GMR24A06 | none | VT033616 | 5 | VT063635 | 2V |
| GMR27G06 | 9,10,11 | VT034258 | 4,13 | VT064563 | 2D,2V |
| GMR29F03 | none | VT037492 | 7,8 | VT065306 | none |
| GMR32A11 | 1,3,4,5,6,7,8,9,10,11,12,13,12V | | | | |

neuron arbors (*Figure 5—figure supplement 1J*). Proboscis mechanosensory neurons also terminate in the same region (*Zhou et al., 2019*). Motor neurons are thus mainly excluded from the base of the compound labial nerve where certain mechanosensory and taste inputs enter the SEZ.

Although bitter taste can cause proboscis retraction (*Dethier, 1976*), we find that bitter inputs do not directly overlap the retractor motor neurons (one example: *Figure 5—figure supplement 1K*). The main retractor motor neuron, mn1, is near peptidergic Hugin neurons, thought to inhibit feeding initiation (*Melcher and Pankratz, 2005*; *Figure 5—figure supplement 1L,M*). Hugin neurons do not

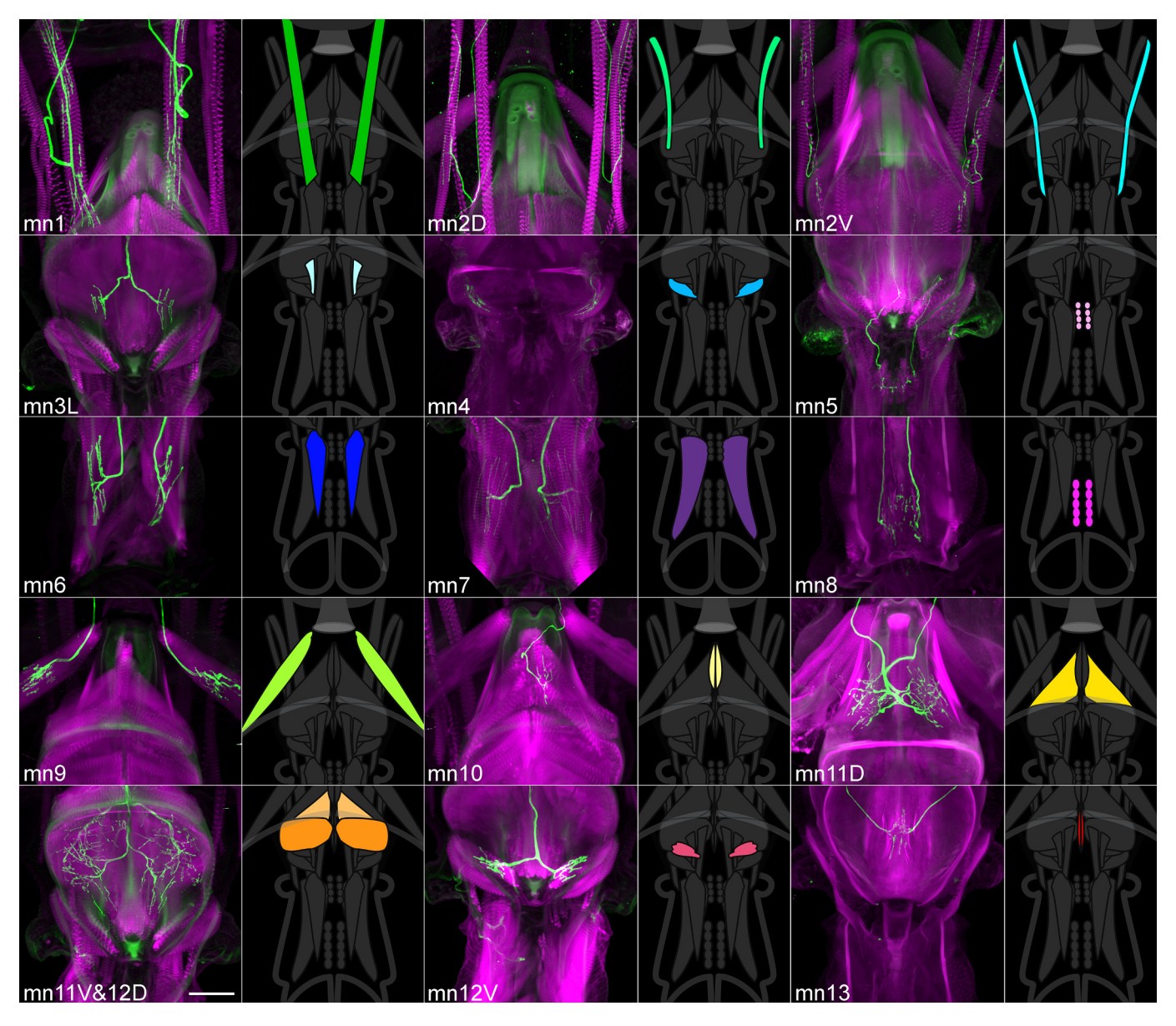

**Figure 4.** A collection of fly strains to genetically control every proboscis muscle. Confocal stacks of split GAL4 lines showing the proboscis muscles (magenta) targeted by the motor neurons of the collection (left images; green). (Note: cuticular structures can also autofluoresce green). Scale bar: 50 μm. Gain and contrast adjusted. Right images: location of those muscles in the head schematic from *Figure 3*, at a reduced scale.

The online version of this article includes the following figure supplement(s) for figure 4:

**Figure supplement 1.** Sparse lines providing genetic access to specific proboscis motor neurons.
**Figure supplement 2.** Completeness of motor neuron coverage in split GAL4 collection.

lie close to 9 or 11D (not shown), but do show proximity to the salivary mn13 (*Figure 5—figure supplement 1N*).

Fdg, a pair of command-like neurons that can activate several features of the feeding motor program (*Flood et al., 2013*), was segmented here from a broader GAL4 line. After computationally aligning images to a common template, Fdg shows little or no overlap with a retractor, protractor, or pharyngeal motor neuron (*Video 12*), suggesting that there are additional interneuron layers yet to be found.

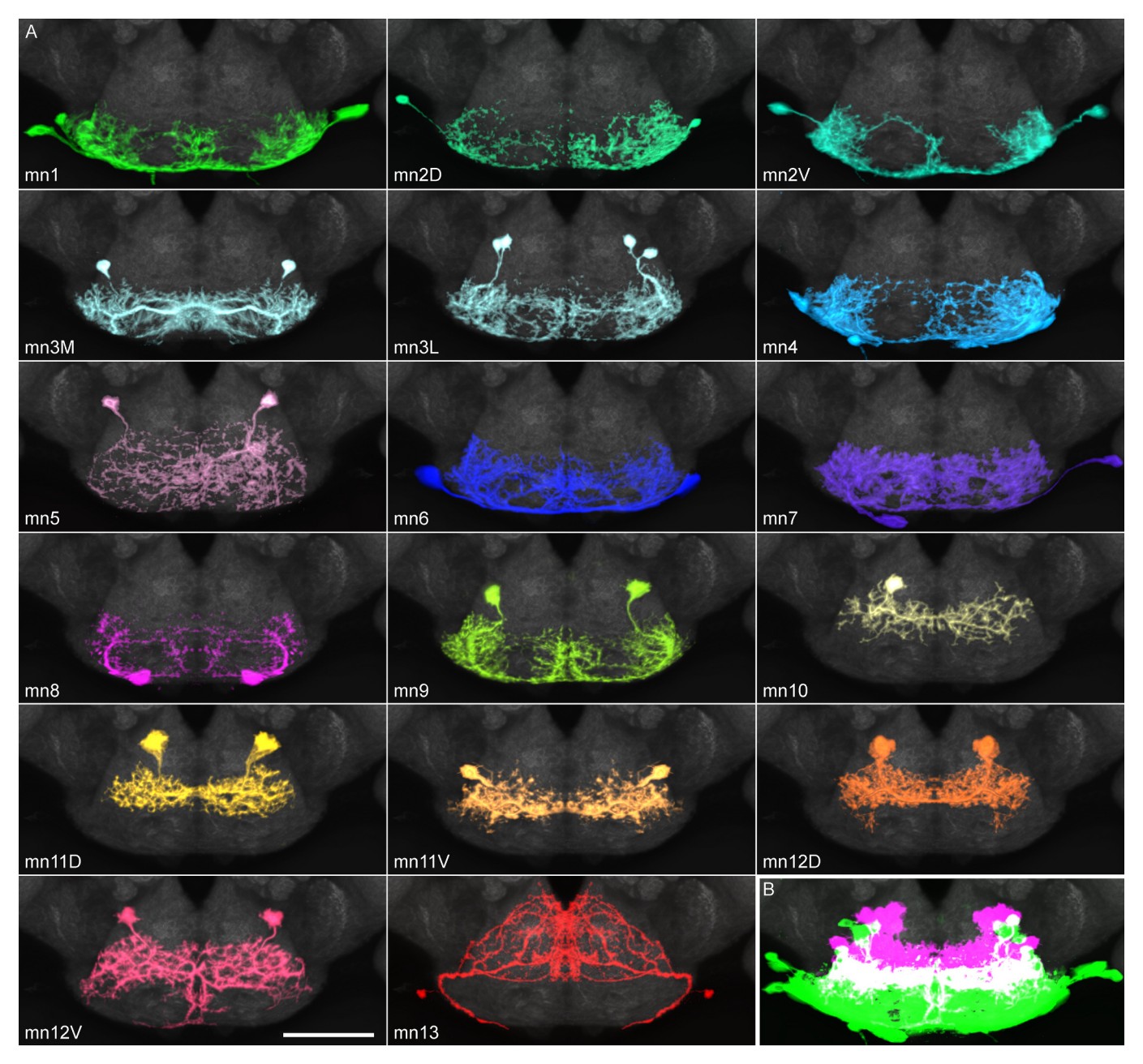

**Figure 5.** Proboscis motor neuron collection: arbors in brain (subesophageal zone). (**A**) Motor neurons from the split GAL4 lines in *Table 1*, segmented to show arbors in isolation. Colors match muscles in previous figures. Most motor neurons are segmented from split GAL4 combinations in which their arbors are clearly distinguishable, with the exception that mn3M, 8, 11V, 12D and 13 are segmented from stochastic staining in order to separate them from nearby cells. Single neurons from stochastic staining are superimposed upon their mirror images to show bilateral arbors, for comparison with the neurons segmented bilaterally. mn10 is shown unilaterally, since it was never found bilaterally in any split GAL4 combination. Scale bar: 50 μm. (**B**) Motor neurons colored according to whether their dendrites are primarily dorsal (magenta) or ventral (green). Magenta: 10, 11D, 11V, 12D, 12V (13: not shown). Rest: ventral.

The online version of this article includes the following figure supplement(s) for figure 5:

**Figure supplement 1.** Location of motor neuron dendrites relative to other cell types in brain.

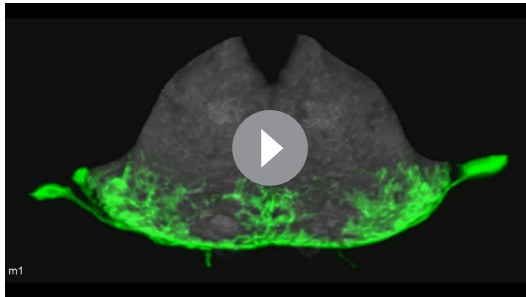

**Video 11.** Each proboscis motor neuron type, isolated using segmentation software. Cropped to show only SEZ at the bottom of brain. Hole at top: esophageal foramen.
https://elifesciences.org/articles/54978#video11

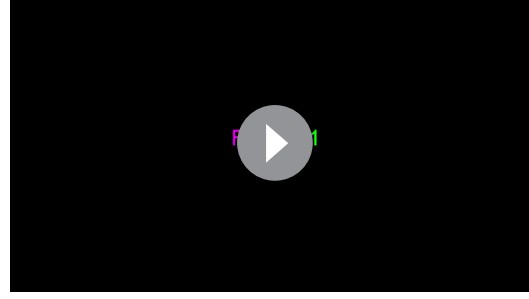

**Video 12.** Translation through computational alignment of feeding command-like neuron 'Fdg' (magenta) with motor neurons 1, 9 and 11D (green) shown sequentially. Motor neurons: confocal stacks. Fdg: manually segmented from a line with a broader expression pattern, NP883-GAL4. Central brain, with SEZ at bottom. Gain and contrast adjusted.
https://elifesciences.org/articles/54978#video12

## Muscles that control the pharynx

Although interneuronal circuits for proboscis motor control will be of future interest, here we aimed to establish the general functions of proboscis muscles. Predictions about function can be made by understanding the insertion sites of muscles. We stained cuticle, tendons and muscle to reveal every muscle insertion site (*Figure 6* and *Figure 6—figure supplement 1*). Previously, three muscles were thought to control the pumping of food through the pharynx *Flood et al., 2013*; *Manzo et al., 2012*; *Tissot et al., 1998*; our staining now reveals a total of eight muscles associated with the pharynx. One (m13) inserts at the junction between the salivary duct and the pharynx (*Figure 6A*) and is predicted to open a valve for salivation. The seven others insert on the pharynx wall (*Figure 6A*), either directly or through tendons (*Figure 6—figure supplement 2A,B*). During feeding, furrows (pseudotrachea) on the labella may channel fluid to the opening of the pharynx (*Figure 6—figure supplement 2C*). Because the seven muscles attached to the pharynx are arrayed along the pharynx in series, we predict that sequential contraction would cause a traveling local dilation of the pharynx wall to move a bolus of fluid through the proboscis (*Video 13*), acting against the elasticity of the pharynx wall rather than antagonistic muscle counterparts. After food has been pumped through the pharynx to the esophagus, a ring of muscles surrounding the

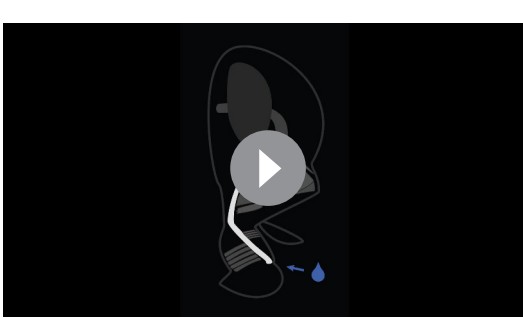

**Video 13.** Predicted mechanism of pumping food (blue) through the pharynx (white) by sequential activation of the seven muscles that insert on the pharynx wall (colored as in *Figures 3* and *4*). Sagittal view of head.
https://elifesciences.org/articles/54978#video13

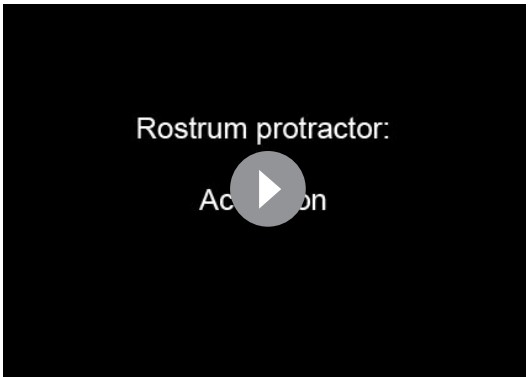

**Video 14.** Behavioral phenotypes of rostrum protractor. First part: activation of rostrum protractor (mn9) with CsChrimson, in frames noted, compared to CsChrimson control, beginning with the proboscis in the resting (retracted) position. Second part: silencing of mn9 with TNT, compared to TNT control, in a feeding assay where normal flies fully extend the proboscis towards a droplet of sucrose. Tethered males, sagittal view (dorsal up), 1/30 speed.
https://elifesciences.org/articles/54978#video14

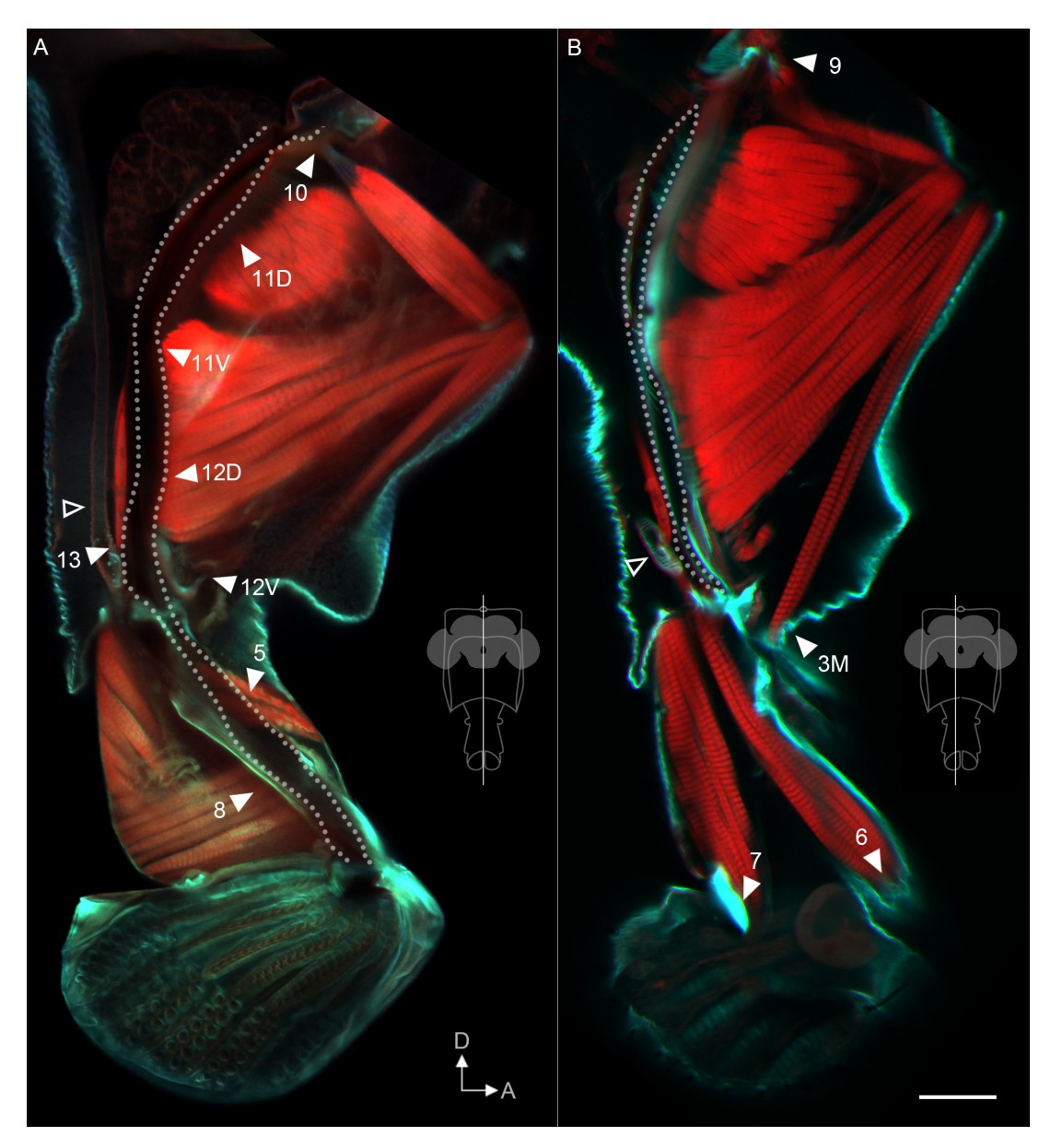

**Figure 6.** Muscle insertion sites predict function. Sagittal view of proboscis, showing phalloidin-stained muscles (red) and calcofluor white-stained cuticle (cyan). Single optical slices from the planes shown in insets, from 300 μm vibratome sections. Dotted line: path of pharynx. Arrowheads: insertion sites of muscles that do contact the pharynx (**A**), and that do not (**B**), predicted to be involved in pumping vs. proboscis positioning, respectively. The salivary muscle 13 can be seen inserting at the junction of the salivary duct (hollow tube – open arrowheads) with the pharynx. 12V is out of the plane of view in (**A**) but its tendon inserts on the pharynx. Rest of muscle insertions shown in *Figure 6—figure supplement 1*. Scale bar: 50 μm.

The online version of this article includes the following figure supplement(s) for figure 6:

**Figure supplement 1.** Additional muscle insertion sites.
**Figure supplement 2.** Features of the alimentary canal.

esophagus suggests that peristaltic action takes over (*Figure 6—figure supplement 2D*). This model, based on our analysis of insertion sites, provides testable predictions for future experiments.

Eight other muscles were of more interest to us as they do not insert on the pharynx (*Figure 6B* and *Figure 6—figure supplement 1*), and are therefore candidates for proboscis positioning. Having genetic access to each motor neuron type, we manipulated each of the eight putative positioning motor neuron types to identify the muscles controlling reaching. We used a comprehensive

battery of tests involving genetic activation and silencing, from the resting proboscis position and also during feeding from an extended position, measuring the position of the proboscis and the angle of each joint.

## Muscles that control rostrum movements

Examining the joints in order of proximal to distal, we first examined the control of the rostrum. Muscle 9 had previously been shown to be involved in rostrum protraction by stochastically targeting its motor neuron from a more complicated expression pattern (*Gordon and Scott, 2009*). Now with reproducible genetic access to the bilateral pair of mn9, we activated it with CsChrimson (*Klapoetke et al., 2014*) and confirmed that it elicited proboscis extension (*Figure 7A,K,L*, *Figure 7—figure supplement 1*). (Joint angle measurements are reported as change in angle from baseline wherever possible, as explained in the Materials and methods, but are also available as raw angles in *Figure 7—figure supplement 2*). We saw not only protraction of the rostrum but also extension of the haustellum (*Figure 7B–C*, *Video 14*). Activation during feeding, when the proboscis is already extended, did not elicit any further extension (*Figure 7D–E*). Silencing mn9 with tetanus toxin (*Sweeney et al., 1995*) did not impair proboscis position or joint angles at rest (*Figure 7F–H*), but prevented the movement of both the rostrum and haustellum during natural feeding (*Figure 7I–J*, *Video 14*), leaving only the movement of the labella (Figure 9H, *Video 14*). mn9 thus shows both activation and silencing phenotypes at the rostrum joint, where its muscle is physically located, but also at the more distant haustellum joint. This demonstrates an example of coupling, where one joint's action affects others, often for biomechanical reasons (*Lang and Schieber, 2004*). In sum, muscle 9 is a protractor of the rostrum, but with wider effects.

What muscles might counteract the protractor action of muscle 9, leading to rostrum retraction? Three muscles (1, 2D, 2V) could act as rostrum retractors based on their origins on the posterior wall of the head, and insertions near the distal end of the rostrum where they could potentially withdraw the rostrum into the head capsule (*Figure 7M–O*). Activation of the motor neurons for the largest of these, muscle 1, indeed retracted the proboscis into the head further

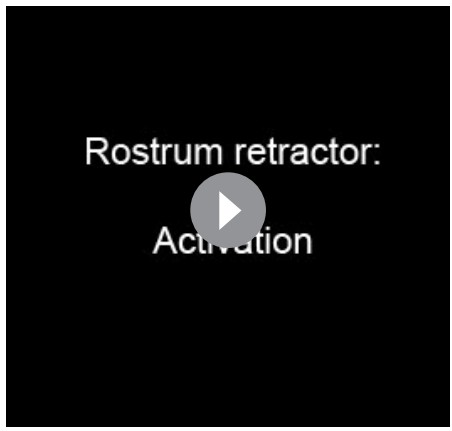

**Video 15.** Behavioral roles of rostrum retractor. First part: activation of the main rostrum retractor (mn1) with CsChrimson, in frames noted, compared to CsChrimson control, beginning with the proboscis in the extended position during feeding. Second part: silencing of mn1 with TNT, compared to TNT control, in the resting proboscis position to demonstrate that mn1 silencing results in incomplete proboscis retraction. Tethered males, sagittal view (dorsal up), 1/30 speed.
https://elifesciences.org/articles/54978#video15

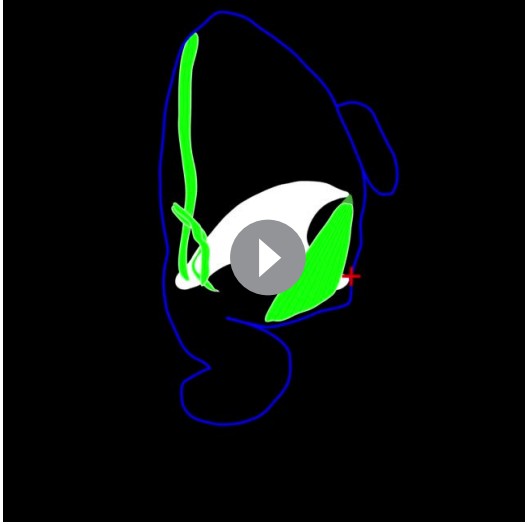

**Video 16.** Mechanism of rostrum movement. Head (blue), muscles involved in rostrum movement (green), and apodeme within rostrum (white). First part: schematic of rostrum movement (pivot point: red crosshairs). Muscles: 1 (long, at left), 2V and 2D (short, at left), and 9 (right). Dorsal up, anterior at right. Second part: same structures segmented from confocal images (sagittal view, maximum projection) with rostrum more retracted (left) or extended (right), showing direction of muscle action (arrows). Third part: same segmented structures in rotating views with rostrum more retracted followed by more extended.
https://elifesciences.org/articles/54978#video16

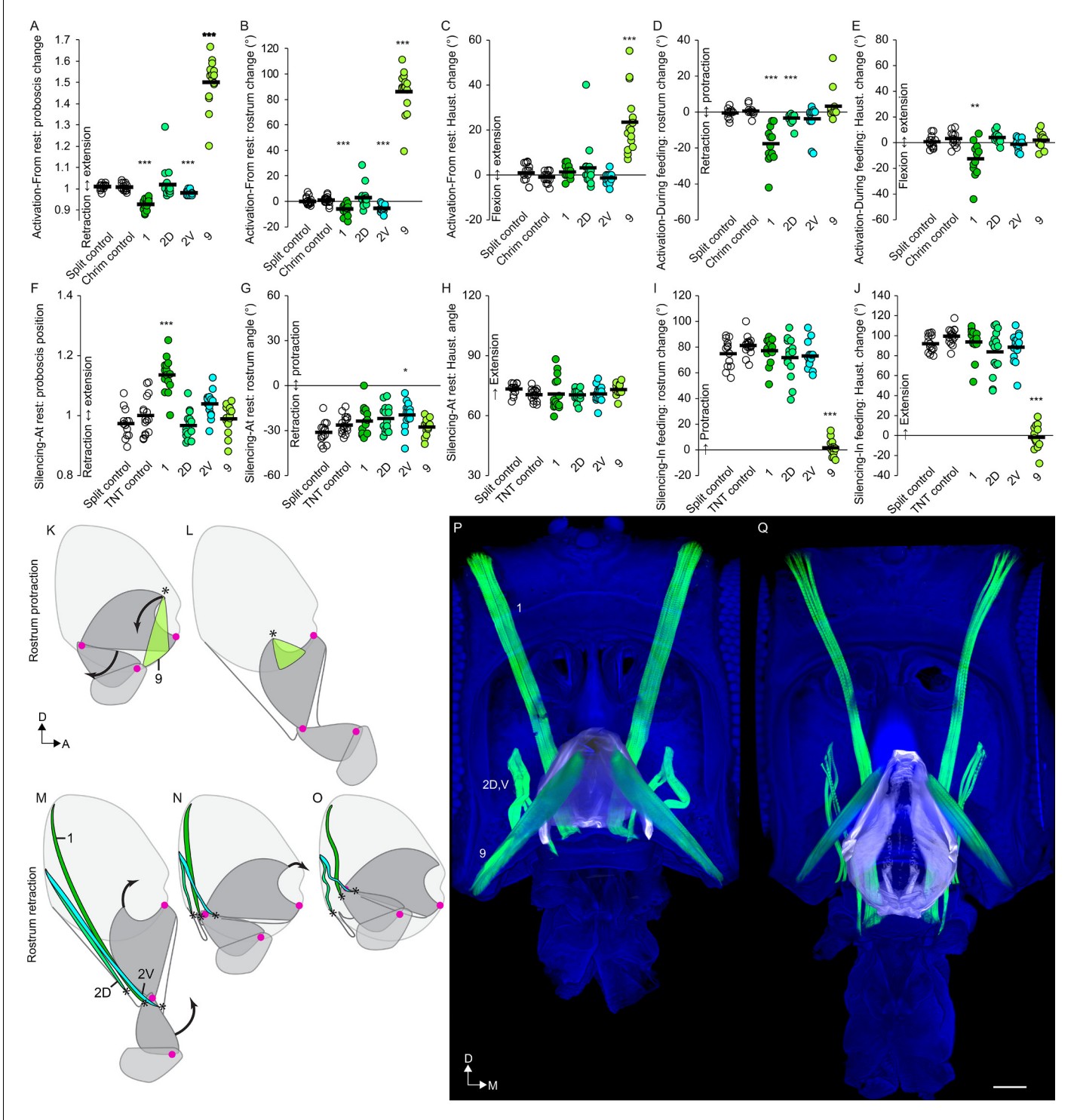

**Figure 7.** Motor control of the rostrum. (**A–E**) CsChrimson activation of split GAL4s for rostrum motor neurons, compared to controls (unfilled), quantifying change in proboscis position from rest (total extension of proboscis at maximum movement divided by at rest; therefore no movement = 1) (**A**), change in rostrum angle from rest (**B**), change in haustellum angle from rest (**C**), change in rostrum angle from a protracted position during feeding (**D**), and change in haustellum angle from an extended position during feeding (**E**). (**F–J**) TNT silencing of split GAL4s for rostrum motor neurons, compared to controls, quantifying proboscis position at rest (**F**), rostrum angle at rest (**G**), haustellum angle at rest (**H**), change in rostrum angle from rest to feeding position (**I**), and change in haustellum angle from rest to feeding position (**J**). Bar: mean. Biological replicates, n = 14–16 flies/genotype. Asterisks: unpaired t-tests, experimental (colored) vs. each control (showing least significant), with multiple testing correction. *p<0.05, **p<0.01, ***p<0.001, no asterisk, not signficant. See Methods for further explanations of metrics shown. (**K–L**) Sagittal schematics of proboscis movements

*Figure 7 continued on next page*

*Figure 7 continued*

controlled by muscle 9 (green): protraction of rostrum and extension of haustellum (arrows). Muscle 9 origin: ventral wall of head. Muscle 9 insertion: internal part of rostrum cuticle (asterisk). Proboscis segments (dark gray) pivot around joints (magenta dots). (**M–O**) Proboscis movements controlled by muscles 1, 2D and 2V (colored): retraction of rostrum and haustellum (arrows). Muscle origins: posterior wall of head. (**P–Q**) Frontal view of whole-mount heads (blue) with segmented muscles (green, numbered) and the apodeme within the rostrum (white) that swings outward during rostrum extension, shown with rostrum retracted (**P**) or extended (**Q**). Scale bar: 50 μm.

The online version of this article includes the following figure supplement(s) for figure 7:

**Figure supplement 1.** Examples of rostrum muscle actions.
**Figure supplement 2.** Raw joint angles, not normalized, for all motor neurons from *Figures 7–9*.

than where it sits at rest (*Figure 7A*), by retraction of the rostrum (*Figure 7B*) but not the haustellum (*Figure 7C*). Activating mn1 while the proboscis was extended for feeding triggered both retraction of the rostrum and flexion of the haustellum (*Figure 7D–E*, *Figure 7—figure supplement 1C–D*, *Video 15*). Whereas activation triggered retraction, silencing mn1 showed the opposite phenotype, impairing the retraction of the proboscis at rest (*Figure 7F*, *Video 15*). Consistent with the effects of muscle 1 on more than one joint in the activation experiments (*Figure 7D–E*), the impaired retraction in silencing experiments could not be attributed to one specific joint (*Figure 7G–H*). In sum, muscle 1 is a retractor, located in the rostrum but with additional effects on the haustellum.

The other two putative rostrum retractor muscles, m2D and m2V, also affected retraction, but more specifically for the rostrum. They showed distinct roles; mn2V activation caused rostrum retraction from rest, whereas mn2D activation caused retraction from the extended position (*Figure 7B, D*). Silencing mn2V caused a small impairment in how retracted the rostrum is at rest (*Figure 7G*). Other silencing phenotypes for mn2V and mn2D were not observed (*Figure 7F–H*), suggesting that they may be at least partly redundant. In sum, muscles 2D and 2V are accessory rostrum retractors. Silencing the retractor muscles, 1, 2D or 2V, did not impair the extension of the rostrum or the haustellum during feeding, thus we conclude that these muscles are not required for proboscis extension (*Figure 7I–J*), only for retraction.

Combining these behavioral results with anatomical results, we propose a model of the action of the rostrum. Imaging the rostrum in different positions shows that its extension correlates with the movement of an apodeme, a large part of cuticle within the rostrum (*Figure 7P–Q*, *Video 16*). Only one point of the apodeme attaches to the external cuticle; this point forms the joint around which the rostrum rotates, seen in *Video 1*. Muscle 9 causes rostrum protraction by anchoring on the head capsule and pulling on the free dorsal arm of the apodeme (via a tendon visible in *Figure 6B*), swinging it ventrally out of the head capsule. To counteract this protraction, retraction is controlled by muscles 1, 2D and 2V, whose dorsal ends anchor on the head capsule, and whose ventral ends insert not on the apodeme, but on cuticle near its ventral end, where their contraction can draw the rostrum back into the head (and potentially aid haustellum retraction, discussed below). Their role in rostrum retraction is supported by their decreased length in the retracted state, their decreased distance between sarcomeres (muscle striations) in the retracted state (consistent with contraction) (*Figure 7P–Q*), and by the behavioral results. In sum, for the rostrum, we find evidence that muscle 9 controls protraction, and that three muscles control retraction (1, 2D, 2V).

## Muscles that control haustellum movements

With the above characterization of rostrum muscles, we then sought to describe the much less well understood haustellum. Although it was affected by rostrum muscles described above, the haustellum can also move independently of the rostrum (*Figure 2M–O*, *Video 4*). We determined that the muscle responsible for haustellum extension is the previously uncharacterized muscle 4; activation of mn4 elicited haustellum extension both from the rest position (*Figure 8C,K,L*, *Figure 8—figure supplement 1A–B*, *Video 17*) and from the extended position during feeding (*Figure 8E*), resulting in an overall change in proboscis position (*Figure 8A*). (Raw angles available in *Figure 7—figure supplement 2*). These activation phenotypes were specific to the haustellum, not affecting the rostrum (*Figure 8B,D*). Silencing instead of activating mn4 caused a defect specific to haustellum extension during feeding (*Figure 8F–J*, *Video 17*). Muscle 4 therefore shows both activation and silencing phenotypes for haustellum extension. Since haustellum extension during feeding was virtually abolished

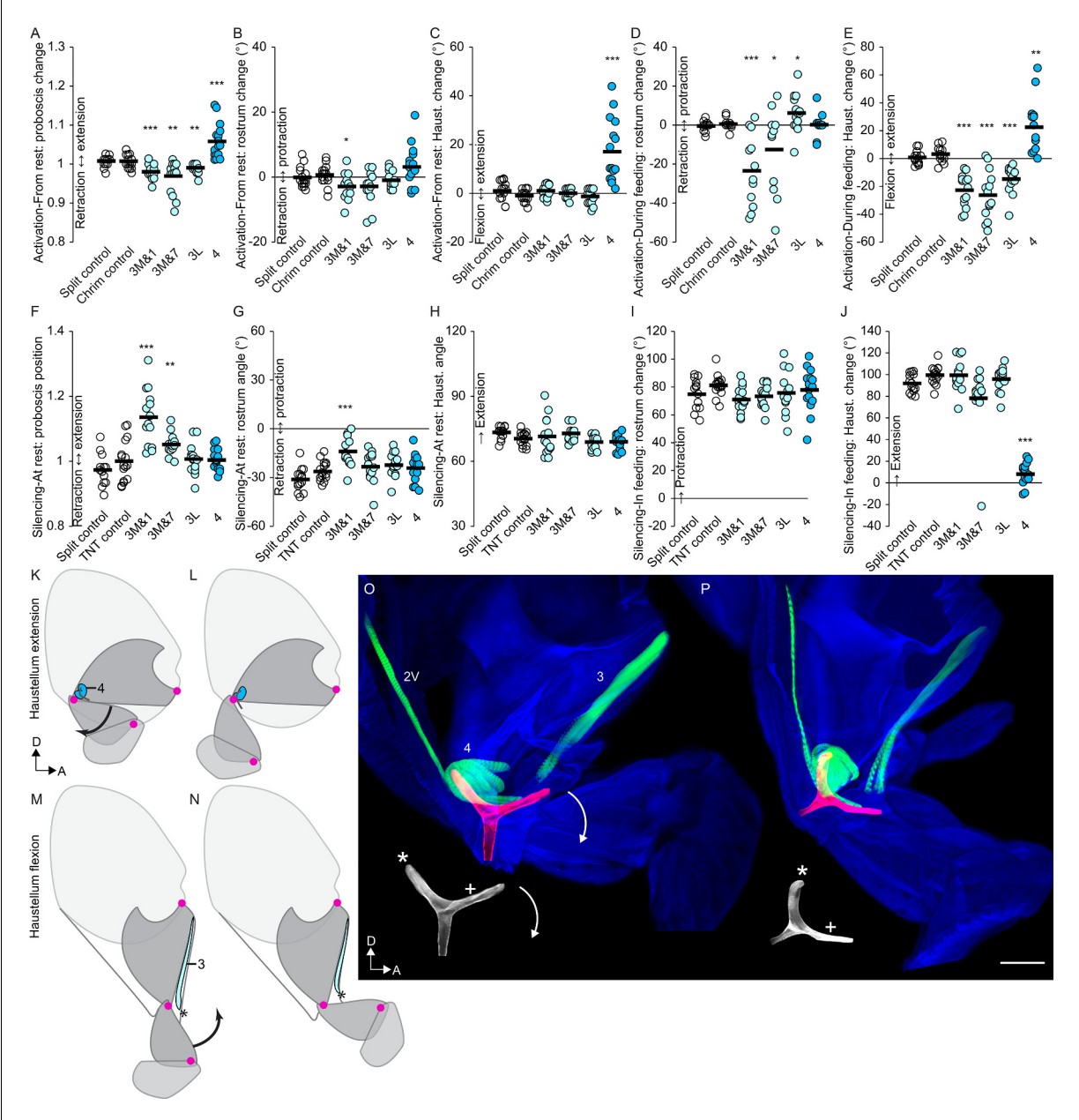

**Figure 8.** Motor control of the haustellum. (**A–E**) CsChrimson activation of haustellum split GAL4s, compared to controls (unfilled), quantifying change in proboscis position from rest (**A**), change in rostrum angle from rest (**B**), change in haustellum angle from rest (**C**), change in rostrum angle from a protracted position during feeding (**D**), and change in haustellum angle from an extended position during feeding (**E**). (**F–J**) TNT silencing of haustellum split GAL4, compared to controls, quantifying proboscis position at rest (**F**), rostrum angle at rest (**G**), haustellum angle at rest (**H**), change in rostrum angle from rest to feeding position (**I**), and change in haustellum angle from rest to feeding position (**J**). Bar: mean. Biological replicates, n = 14–16 flies/genotype. Asterisks: unpaired t-tests, experimental (colored) vs. each control (showing least significant), with multiple testing correction. *p<0.05, **p<0.01, ***p<0.001, no asterisk, not signficant. (**K–L**) Sagittal schematics of proboscis movement controlled by muscle 4 (blue): extension of haustellum (arrow). Muscle 4 location: near haustellum joint. Proboscis segments (dark gray) pivot around joints (magenta dots). (**M–N**) Proboscis movement controlled by muscle 3 (light blue): flexion of haustellum (arrow). Muscle 3 origin: anterior rostrum. Insertion: asterisk. (**O–P**) Haustellum mechanism. Lateral view of proboscis (blue) with segmented muscles (green, numbered) and a Y-shaped apodeme (red in composite, white below), from thick sections with haustellum partly flexed (**O**) or partly extended (**P**). Muscle 4 inserts on the free dorsal arm of the apodeme (asterisk). Muscle 2V inserts on the anterior apodeme arm (plus sign). Muscle 3 inserts in the haustellum (via tendons not stained here). Arrows: rotation of apodeme, controlling extension of haustellum. Scale bar: 50 μm.

The online version of this article includes the following figure supplement(s) for figure 8:

**Figure supplement 1.** Examples of haustellum muscle actions.

by silencing mn4, its muscle may be the only one required for haustellum extension. Although silencing mn9 also abolished haustellum extension, that may be because it also blocks rostrum extension, making haustellum extension more difficult mechanically.

In opposition to extension, we find that haustellum flexion is controlled by muscle 3. Before behavioral testing, we stained 139 different combinations of split GAL4 lines to attempt to target mn3 (data not shown). None cleanly targeted motor neurons for all fibers of muscle 3, but its lateral fibers and medial fibers could be separately targeted (*Table 1*). (Muscles are often made up of multiple fibers; when only a single primary motor neuron controls that muscle, it branches to boutons on each muscle fiber. We describe these lateral and medial fibers of m3 as one muscle since all have adjacent insertions and origins). One split GAL4 line targeted the lateral fibers (mn3L) cleanly, with no other proboscis motor neurons present. No lines cleanly targeted motor neurons for the medial fibers, thus we selected two imperfect lines, each of which targeted one additional proboscis motor neuron type, 'mn3M&1' and 'mn3M&7'. The function of muscle 3M could be inferred if a phenotype were found in both these lines, and not in both of the clean lines for the contaminating cell types (1 or 7).

Activating mn3 demonstrated that it acts as a haustellum flexor. mn3M&1, mn3M&7 and mn3L all triggered haustellum flexion during feeding (*Figure 8E,M,N*, *Figure 8—figure supplement 1C–D*, *Video 18*). Although the contaminating cell type mn1 does so as well (*Figure 7E*), the contaminating mn7 does not (Figure 9N), suggesting that muscle 3 is responsible. From the resting proboscis position, all mn3 lines could trigger additional proboscis retraction (*Figure 8A*), although that could not be attributed to a specific joint (*Figure 8B–C*). As opposed to the effects on the haustellum, activation did not produce consistent effects on the rostrum (*Figure 8D*).

In silencing experiments, mn3M&1 and mn3M&7 both impaired proboscis retraction at rest (*Figure 8F*, *Video 18*). Although the contaminating cell type mn1 does so as well (*Figure 7F*), the contaminating mn7 does not (*Figure 9O*), suggesting that muscle 3 is responsible. Silencing mn3 lines otherwise did not produce consistent phenotypes (*Figure 8F–J*), possibly because mn3L and mn3M were not both targeted together. In sum, by examining three different lines for mn3, we conclude that muscle 3 is a haustellum flexor.

Combining these behavioral results with anatomical results, we propose a model of the action of the haustellum. Imaging the haustellum in different positions (*Figure 8O–P*) shows that its extension

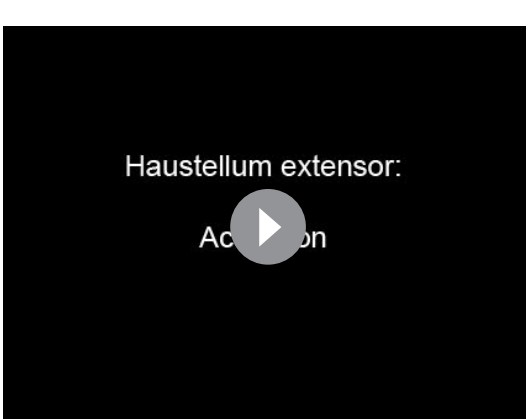

**Video 17.** Behavioral roles of haustellum extensor. First part: activation of haustellum extensor (mn4) with CsChrimson, in frames noted, compared to CsChrimson control, beginning with the proboscis in the resting (retracted) position. Second part: silencing of mn4 with TNT, compared to TNT control, in a feeding assay where normal flies fully extend the proboscis towards a droplet of sucrose. Tethered males, sagittal view (dorsal up), 1/30 speed.
https://elifesciences.org/articles/54978#video17

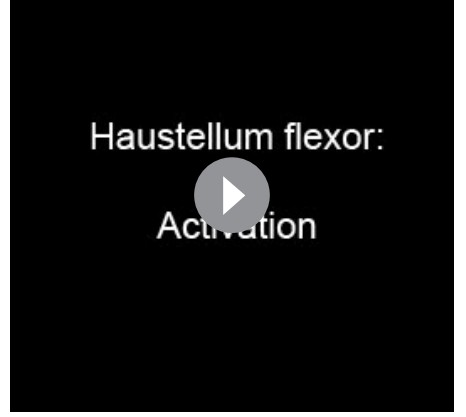

**Video 18.** Behavioral roles of haustellum flexor. First part: activation of one of the two m3 haustellum flexors (using line mn3M&7) with CsChrimson, in frames noted, compared to CsChrimson control, beginning with the proboscis in the extended position during feeding. Second part: silencing of mn3M&7 with TNT, compared to TNT control, in the resting proboscis position to demonstrate that mn3M&7 silencing results in incomplete proboscis retraction. Tethered males, sagittal view (dorsal up), 1/30 speed.
https://elifesciences.org/articles/54978#video18

correlates with the rotation of a Y-shaped apodeme found within the joint. Of the three arms of the Y, the ventroposterior arm was continuous with the external cuticle of the rostrum, the anterior arm was continuous with the external cuticle of the haustellum, and the dorsal arm was free. We propose that a muscle pulling against the free dorsal arm could rotate the haustellum anteriorly away from the rostrum. Indeed, we found that muscle 4 originates from a rigid part of the rostrum and inserts upon the free dorsal arm of the Y, where its contraction would cause rotation of the apodeme and thus haustellum extension. Behavioral experiments above did show that muscle 4 was a haustellum extensor.

Another muscle, 2V, inserts upon the anterior arm (*Figure 8O–P*) where it could rotate the apodeme back in the other direction, dorsally. This muscle showed no haustellum phenotype, but its insertion site is suggestive enough to bear further future investigation for a role in haustellum flexion, possibly masked here by muscle redundancy. (Not included in *Figure 8O–P* is muscle 1, because although it had an activation phenotype on the haustellum (*Figure 7E*), this was possibly an indirect effect such as coupling, since there were no silencing effects and the muscle is not located in a position to control haustellum flexion). Instead, another muscle, 3, was ideally located to act as a flexor (*Figure 8O–P*); coming from the anterior rostrum, it crosses the joint with the haustellum so that it could pull on the haustellum to fold it in towards the rostrum. In behavioral experiments (above), muscle 3 indeed triggered haustellum flexion. In sum, we find evidence that muscle 4 controls haustellum extension and that muscle 3 (possibly also 2V) controls flexion. Indeed, based on distances between sarcomeres (*Figure 8O–P*), muscle 4 appears more contracted when the haustellum is more extended, and 2V and 3 appear more contracted when the haustellum is more flexed.

## Muscles that control labellar movements

The remaining two muscles (6 and 7) of the eight tested for roles in proboscis positioning were both found to insert upon on the lip-like labella (*Figures 6B* and *9A–C*). Muscle 6 shows both activation and silencing phenotypes for labellar extension (*Figure 9F,H*, *Video 19*). Muscle 7 shows activation and silencing phenotypes for labellar abduction (the opening of the two lobes, *Figure 9G,I*, *Video 20*). Although mn7 activation also induced labellar extension (*Figure 9F*), that may be a side effect of abduction; muscle 7 showed no effect on labellar extension when silenced (*Figure 9H*). The line contaminated with mn7, mn3M&7, showed similar phenotypes to mn7 but at lower penetrance (*Figure 9F–I*), likely due to the fact that mn7 was not labeled in all flies of that line (*Table 1*). Muscle 6 and 7 are therefore the only muscles that showed labellar phenotypes. They showed certain effects on other proboscis movements (*Figure 9J–S*) which we interpret to be side effects, since they insert far from the rostrum or haustellum joints (*Figure 9A*). (Raw angles available in *Figure 7—figure supplement 2*).

To summarize the behavioral results, all eight of the muscles predicted to be involved in proboscis positioning (based on the location of their insertion sites) indeed showed positioning phenotypes; six control reaching movements and two control the labella. Most of these findings (*Table 3*) were not previously known or, in some cases, differed from previous work (*Schwarz et al., 2017*).

# Discussion

The present work describes the anatomy and function of motor neurons controlling each muscle of the fly proboscis. Building on previous studies in insects that have analyzed motor neurons in body parts such as the legs and proboscis (*Baek and Mann, 2009*; *Brierley et al., 2012*; *Maniates-Selvin et al., 2020*; *Rajashekhar and Singh, 1994*; *Schwarz et al., 2017*; *Trimarchi and Schneiderman, 1994*), we now provide the first comprehensive, genetically reproducible control of every primary motor neuron type for an appendage. We characterize the motor neurons' outputs to musculature in the periphery and their dendrites in the brain.

Detailed imaging of muscle insertion sites is used to predict that eight muscles participate in proboscis positioning, confirmed in behavioral experiments. With these tools, we produce a model of the function of the musculature and control of each proboscis joint, where a small number of motor neurons can produce a goal-directed reaching behavior.

We consider the neurons of this collection to be primary excitatory motor neurons, as they do not resemble known modulatory motor neurons in adult flies, which are characterized by thin axons and small boutons on multiple muscles (*Rivlin et al., 2004*). Likewise, they do not resemble inhibitory

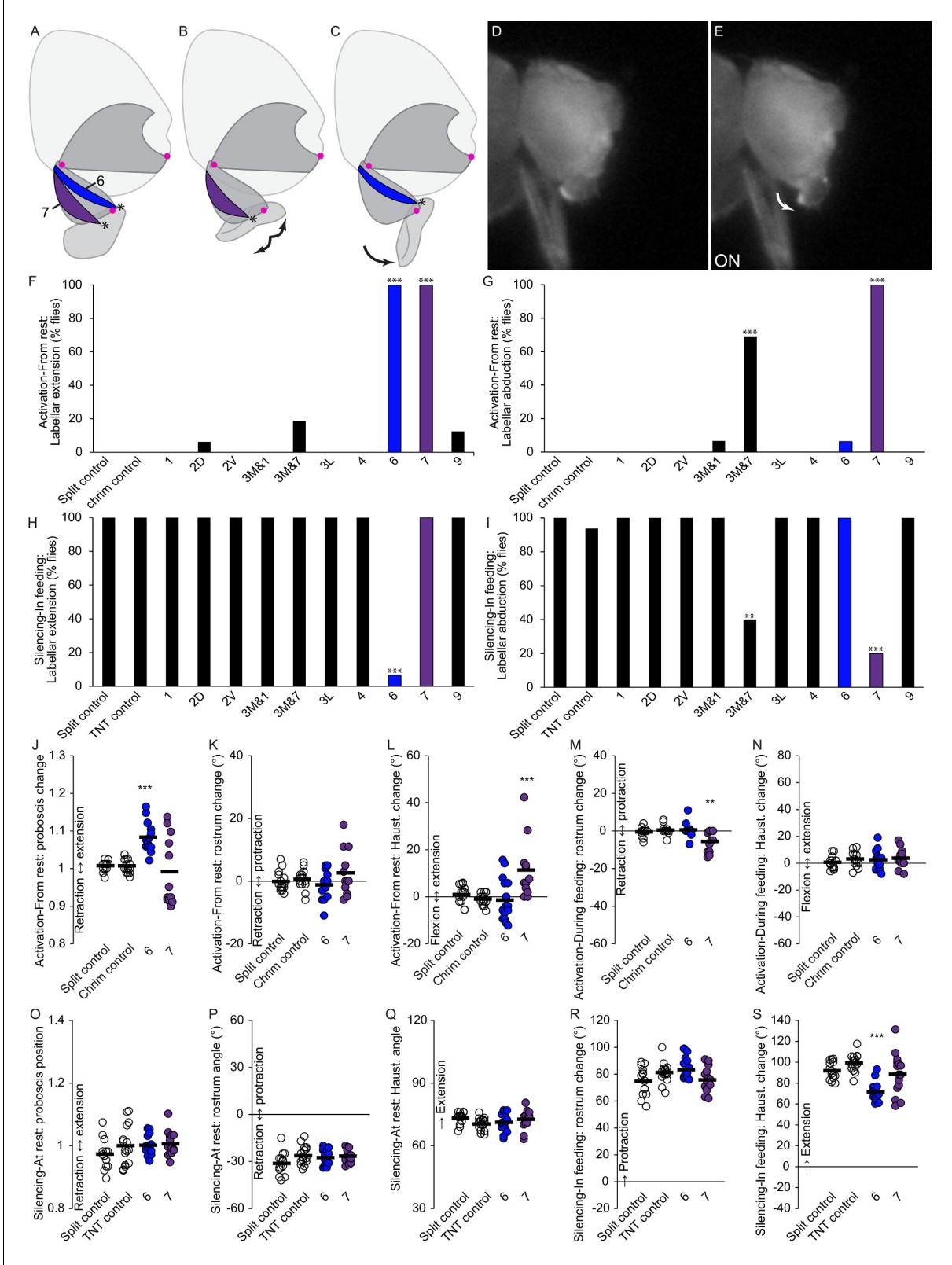

**Figure 9.** Motor control of the labella. (**A–C**) Sagittal schematics of labellar movements controlled by muscles 7 and 6 (colored): abduction (**B**) and extension (**C**) of labella (arrows). Muscle origins: dorsal haustellum. Insertions: at labella (asterisks). Proboscis segments (dark gray) pivot around joints (magenta dots). (**D–E**) Example phenotype: extension of labella in response to CsChrimson activation of mn6 (**E**), compared to resting proboscis before the stimulus (**D**). (**F–I**) % flies showing labellar extension (**F,H**) or abduction (**G,I**) in response to CsChrimson activation (**F,G**) or TNT silencing (**H,I**) of

*Figure 9 continued*

motor neurons listed, compared to controls. Fisher's exact test. Biological replicates, n = 16 flies/genotype. (J–N) CsChrimson activation of split GAL4s listed compared to controls (unfilled), quantifying change in proboscis position from rest (J), change in rostrum angle from rest (K), change in haustellum angle from rest (L), change in rostrum angle from a protracted position during feeding (M), and change in haustellum angle from an extended position during feeding (N). (O–S) TNT silencing of split GAL4s listed compared to controls, quantifying proboscis position at rest (O), rostrum angle at rest (P), haustellum angle at rest (Q), change in rostrum angle from rest to feeding position (R), and change in haustellum angle from rest to feeding position (S). Bar: mean. Biological replicates, n = 14–16 flies/genotype. Asterisks: unpaired t-tests, experimental (colored) vs. each control (showing least significant), with multiple testing correction. *p<0.05, **p<0.01, ***p<0.001, no asterisk, not signficant.

motor neurons found in several arthropods, which mostly innervate multiple muscles (*Wolf, 2014*). Anatomical and behavioral evidence supports the identity of these neurons as primary motor neurons.

The split GAL4 combinations used here to target motor neurons were nicely sparse but not perfect, often containing some neurons in other brain regions. However, they are quite specific in the SEZ, where we think much of the motor circuitry of the proboscis will be found. Triple intersections with SEZ-specific reagents based on Hox expression (*Simpson, 2016*) could be used for further restricting expression. Future work must also examine the cooperative action of multiple motor neurons, rather than just their effects in isolation.

Computational brain alignment suggests that motor neuron dendrites are not found in close proximity to several known cell types involved in feeding. Unknown circuit layers between sensory and motor neurons likely aid with functions such as pattern generation or command-like roles. Although certain mechanosensory neurons have been investigated in the proboscis (*Zhou et al., 2019*), the role of sensory feedback in proboscis reaching remains to be explored; the approach used here could also be applied to the isolation of proprioceptive neurons.

With this motor neuron collection, we provide a description of the muscles that act at each proboscis joint. It extends previous efforts (*Schwarz et al., 2017*) by using a stronger neuronal silencer to find silencing phenotypes for more cell types (*Table 3*), tissue clearing techniques that show more neurons than were thought to be present in GAL4 lines (*Table 2*), and split GAL4 lines for more specificity. While the kinematics of proboscis reaching remains to be explored, the

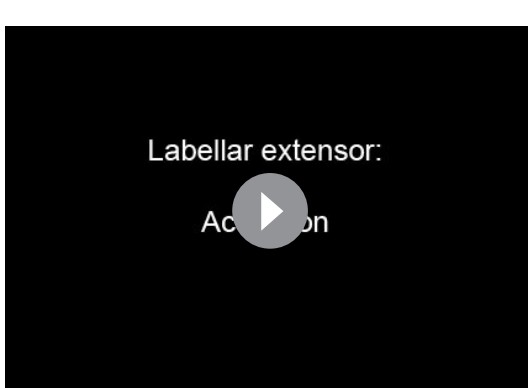

**Video 19.** Behavioral roles of labellar extensor. First part: activation of labellar extensor (mn6) with CsChrimson, in frames noted, compared to CsChrimson control, beginning with the proboscis in the resting (retracted) position. Second part: silencing of mn6 with TNT, compared to TNT control, in a feeding assay. Labella marked in blue in certain frames to show difference in labellar angle. Tethered males, sagittal view (dorsal: up), 1/30 speed.
https://elifesciences.org/articles/54978#video19

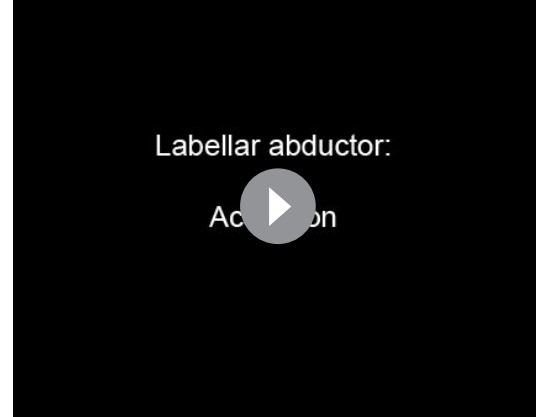

**Video 20.** Behavioral roles of labellar abductor. First part: activation of labellar abductor (mn7) with CsChrimson, in frames noted, compared to CsChrimson control, beginning from the resting position (proboscis retracted, labella closed). Labella marked in blue in certain frames. Second part: silencing of mn7 with TNT, compared to TNT control, in a feeding assay where normal flies open the labella towards a droplet of sucrose. Tethered males, anterior view (dorsal up), 1/30 speed.
https://elifesciences.org/articles/54978#video20

**Table 3.** Summary of motor neuron phenotypes from activation and silencing experiments (also informed by insertion site information). Asterisks: findings that were previously undescribed, or that differ from a previous study (*Schwarz et al., 2017*).

| mn | Activation phenotypes | Silencing phenotypes |
|---|---|---|
| 1 | proboscis retraction | *impaired proboscis retraction |
| 2D | *rostrum retraction from extended | no phenotype (*expected redundancy*) |
| 2V | *rostrum retraction from rest | *impaired rostrum retraction |
| 3 | *haustellum flexion | *impaired proboscis retraction |
| 4 | *haustellum extension | *impaired haustellum extension |
| 6 | labellar extension | impaired labellar extension |
| 7 | *labellar abduction | *impaired labellar abduction |
| 9 | rostrum protract. and *haust. exten. | impaired rostrum protract. and *haust. exten. |

comprehensive genetic tools we describe provide the means to do so.

We discover unexpected complexity of phenotypes from rostrum motor neurons. Both mn9 and mn1 are located near the rostrum, not in a position to control the haustellum directly, yet showed phenotypic effects on the haustellum as well. These may represent examples of biomechanical coupling. We were struck by how common 'side effects' like this were – another example is that, although mn7 is responsible for labellar abduction (with both activation and silencing phenotypes), its activation also triggers labellar extension. We speculate that optogenetically forcing the movement of one body part may cause a fly to move another in response, even if the activated motor neuron is not directly involved in the latter at all, or that forced movement about a joint may cause afferent feedback that leads to activation of other motor neurons. Intrinsic biomechanical properties, such as passive joint forces, are also known to provide non-neural (and rapid) aid during motor responses to mechanical perturbations, for example in cockroach, locust and stick insect legs during locomotion (*Ache and Matheson, 2013*; *Jindrich and Full, 2002*).

These possible side effects of optogenetic activation highlight the utility of combining activation and silencing data to deduce muscle roles, since both methods can have pitfalls (*Yoshihara and Yoshihara, 2018*). By incorporating activation, silencing, and insertion site data, we derive interpretations of muscle roles (*Table 3*) that differ in several respects from those of a previous work based mainly on activation (*Schwarz et al., 2017*).

It will be of interest to determine how flexibility is achieved in the recruitment of proboscis muscles. Several muscles were innervated by only a single primary motor neuron, but in the case where two were present, it is possible that they control slow twitch and fast twitch fibers, as found in both the fly (*Azevedo et al., 2019*) and in mammalian muscle, to fine-tune the strength of a muscle response. Regardless, the dramatic numerical simplicity of only one or two primary motor neurons per muscle in flies presents an experimental advantage for future circuit studies. Given that the proboscis has a varied repertoire of movements in feeding and courtship (*Manzo et al., 2012*; *McKellar et al., 2019*), one exciting avenue would be to decipher the circuit logic of how different movements with multiple behavioral functions can arise from these few motor neurons. Activation and inactivation experiments of individual proboscis motor neuron types support a model of separable control, enabling articulated movements that allow flexibility.

Directed reaching occurs in different forms: humans show sophisticated, learned arm and hand movements (*Parker, 1974*), yet newborns show a more basic, innate form of intentional reaching (*Bower et al., 1970*). In primates, work on the neural coding of goal-directed behavior has often focused on how its features might be encoded by firing properties of single neurons or single brain areas, particularly within the cerebral cortex (*Kalaska, 2019*). Forms of reaching exist in smaller animals such as rodents and cephalopods (*Gutnick et al., 2011*; *Peterson, 1934*). We propose that proboscis reaching in flies could present a new model for understanding the neural circuit basis of goal-directed movement. *Drosophila* has rich behavior, a smaller nervous system, and powerful genetic tools. Additionally, a whole-brain synaptic connectome is underway (see flywire.ai, based on the images of *Zheng et al., 2017*) that can help map how diverse interneurons may converge upon

this small set of motor neurons to produce a repertoire of flexible motor behaviors from a single appendage.

# Materials and methods

## Key resources table

| Reagent type (species) or resource | Designation | Source or reference | Identifiers | Additional information |
|---|---|---|---|---|
| Antibody | mouse mAb anti-bruchpilot (nc82) | Developmental Studies Hybridoma Bank | Cat#nc82; RRID:AB_2314865 | (1:50) |
| Antibody | rabbit anti-GFP | Thermo Fisher Scientific | Cat#A11122; RRID:AB_221569 | (1:500) |
| Antibody | rabbit anti-GFP | Thermo Fisher Scientific | Cat#A10262; RRID:AB_2534023 | (1:1000) |
| Antibody | rat mAb anti-FLAG | Novus Biologicals | Cat# NBP1-06712; RRID:AB_1625981 | (1:200) |
| Antibody | rabbit anti-HA | Cell Signaling Technology | Cat# 3724S; RRID:AB_1549585 | (1:300) |
| Antibody | mouse anti-V5 | AbD Serotec | Cat# MCA1360; RRID:AB_322378 | (1:300) |
| Antibody | goat anti-rabbit AlexaFluor-488 | Thermo Fisher Scientific | Cat#A11034; RRID:AB_2576217 | (1:500) |
| Antibody | goat anti-chicken AlexaFluor-488 | Thermo Fisher Scientific | Cat#A11039; RRID:AB_142924 | (1:500) |
| Antibody | goat anti-mouse AlexaFluor-488 | Thermo Fisher Scientific | Cat #A11001; RRID:AB_2534069 | (1:500) |
| Antibody | goat anti-mouse AlexaFluor-568 | Thermo Fisher Scientific | Cat#A11031; RRID:AB_144696 | (1:500) |
| Antibody | goat anti-rabbit Alexa Fluor-488 | Thermo Fisher Scientific | Cat#A32731; RRID:AB_2633280 | (1:1000) |
| Antibody | goat anti-mouse Cy3 | Jackson Immunoresearch | Cat#115-166-003; RRID:AB_2338699 | (1:250) |
| Antibody | goat anti-rat AlexaFluor-568 | Thermo Fisher Scientific | Cat #A11077; RRID:AB_2534121 | (1:500) |
| Antibody | goat anti-rat AlexaFluor-633 | Thermo Fisher Scientific | Cat#A21094; RRID:AB_141553 | (1:500) |
| Chemical compound, drug | Texas Red-X Phalloidin | Life Technologies | Cat#T7471 | |
| Chemical compound, drug | Calcofluor White | Sigma-Aldrich | Cat#F3543 | |
| Chemical compound, drug | Congo Red | Sigma-Aldrich | Cat#C676 | |

*Continued on next page*

*Continued*

| Reagent type (species) or resource | Designation | Source or reference | Identifiers | Additional information |
|---|---|---|---|---|
| Chemical compound, drug | Alexa Fluor 633 Phalloidin | Life Technologies | Cat#A22284 | |
| Chemical compound, drug | All-trans retinal | Toronto Research Chemical | Cat# R240000 | |
| Genetic reagent *Drosophila melanogaster* | Canton S | Bloomington Stock Center | RRID:BDSC_64349 | |
| Genetic reagent (*D. melanogaster*) | Rubin and VT GAL4 lines listed in *Table 2* | *Jenett et al., 2012*; *Pfeiffer et al., 2010*; *Tirian and Dickson, 2017* | N/A | |
| Genetic reagent (*D. melanogaster*) | Split GAL4 lines targeting proboscis muscles, listed in *Table 1* | This paper | N/A | Split GAL halves from G. Rubin and B. Dickson (*Jenett et al., 2012*; *Pfeiffer et al., 2010*; *Tirian and Dickson, 2017*) |
| Genetic reagent (*D. melanogaster*) | 10XUAS-IVS-mCD8::GFP in su(Hw)attP5 (pJFRC2) | Gerald Rubin (*Pfeiffer et al., 2010*) | N/A | |
| Genetic reagent (*D. melanogaster*) | 20XUAS-CsChrimson-mCherry in su(Hw)attP5 | Insertion from Vivek Jayaraman, construct from *Klapoetke et al., 2014* | N/A | |
| Genetic reagent (*D. melanogaster*) | UAS-TeTxLC.TNT | *Sweeney et al., 1995* | N/A | |
| Genetic reagent (*D. melanogaster*) | pBPhsFlp2::PEST in attP3;; pJFRC201-10XUAS-FRT > STOP > FRT-myr::smGFP-HA in VK00005 | *Nern et al., 2015* | N/A | |
| Genetic reagent (*D. melanogaster*) | pJFRC240-10XUAS-FRT > STOP > FRT-myr::sm GFP-V5-THS-10 XUAS-FRT > STOP > FRT-myr::sm GFP-FLAG in su(Hw)attP1 | *Nern et al., 2015* | N/A | |
| Genetic reagent ' (*D. melanogaster*) | VT017251-LexA | *Hampel et al., 2017* | N/A | |
| Genetic reagent (*D. melanogaster*) | HUGS3-GAL4 | *Melcher and Pankratz, 2005* | N/A | |
| Genetic reagent (*D. melanogaster*) | Gr64f-GAL4 [737-5];Gr64f-GAL4 [737-1] | *Dahanukar et al., 2007* | N/A | |
| Genetic reagent (*D. melanogaster*) | Gr66a-GAL4 (II) | *Dunipace et al., 2001* | N/A | |
| Genetic reagent (*D. melanogaster*) | NP883-GAL4 | *Flood et al., 2013* | N/A | |

*Continued on next page*

*Continued*

| Reagent type (species) or resource | Designation | Source or reference | Identifiers | Additional information |
|---|---|---|---|---|
| Genetic reagent (*D. melanogaster*) | NP5137-GAL4 | *Flood et al., 2013* | N/A | |
| Software, algorithm | Adobe Photoshop | Adobe Systems (www.adobe.com/products/photoshop.html) | RRID:SCR_014199 | |
| Software, algorithm | Adobe Illustrator | Adobe Systems (https://www.adobe.com/products/illustrator.html) | RRID:SCR_010279 | |
| Software, algorithm | Computational Morphometry Toolkit | *Jefferis et al., 2007* | RRID:SCR_002234 | |
| Software, algorithm | LabView | National Instruments (www.labview.com) | RRID:SCR_014325 | |
| Software, algorithm | ImageJ | https://imagej.nih.gov/ij/ | RRID:SCR_003070 | |
| Software, algorithm | Icy | http://icy.bioimageanalysis.org/ | RRID:SCR_010587 | |
| Software, algorithm | gVision | Gus Lott (http://gvision-hhmi.sourceforge.net/) | N/A | |
| Software, algorithm | G*Power | http://www.gpower.hhu.de/ | RRID:SCR_013726 | |
| Software, algorithm | VVDViewer | https://github.com/takashi310/VVD_Viewer | N/A | |

## Terminology

The area formerly known as the sub(o)esophageal ganglion (SOG/SEG) is now called the gnathal ganglion (*Ito et al., 2014*). The gnathal ganglion, however, does not encompass the whole area ventral to the esophageal foramen, the subesophageal zone (SEZ). We use the term SEZ because the motor neurons described here are not confined to the gnathal ganglion, but are all within the larger area of the SEZ.

The axis system used is that of the body axis (*Ito et al., 2014*). During feeding the proboscis mostly extends ventrally toward the surface the fly is standing on, so the head is dorsal, the tip of the proboscis ventral, with the maxillary palps on the anterior surface.

## Fly stocks and rearing

The GAL4-UAS system was used to direct gene expression to defined populations of cells (*Brand and Perrimon, 1993*). The split GAL4 system (*Luan et al., 2006*; *Pfeiffer et al., 2010*) was used to express the activation domain (AD) and the DNA-binding domain (DBD) of GAL4 under the separate control of two genomic enhancers, to obtain the intersection of their expression patterns, with the goal of obtaining a sparser pattern. GAL4, split GAL4 and LexA stocks were gifts of Gerald Rubin and Barry Dickson (*Jenett et al., 2012*; *Pfeiffer et al., 2010*; *Tirian and Dickson, 2017*). GAL4s and DBDs were inserted into the landing site attP2; ADs and LexAs were inserted into attP40. Other lines used were: wildtype: Canton S 'A' (a gift of Jeff Hall), GFP: *pJFRC2-10x-UAS-IVS-mCD8::GFP in su(Hw)attP5* (*Pfeiffer et al., 2010*), CsChrimson: *20XUAS-CsChrimson-mCherry*

(*Klapoetke et al., 2014*) in *su(Hw)attP5*, TNT: *UAS-TeTxLC.TNT* (*Sweeney et al., 1995*), eye mechanosensory neurons: *VT017251-LexA* (*Hampel et al., 2017*), Hugin: *HUGS3-GAL4* (*Melcher and Pankratz, 2005*), sweet taste neurons: *Gr64f-GAL4 [737-5];Gr64f-GAL4 [737-1]* (*Dahanukar et al., 2007*), bitter taste neurons: *Gr66a-GAL4 (II)* (*Dunipace et al., 2001*), Fdg neurons: *NP883-GAL4* and *NP5137-GAL4* (*Flood et al., 2013*), multicolor flipout stochastic labelling: *pBPhsFlp2::PEST in attP3;; pJFRC201-10XUAS-FRT > STOP > FRT-myr::smGFP-HA in VK00005, pJFRC240-10XUAS-FRT > STOP > FRT-myr::smGFP-V5-THS-10XUAS-FRT > STOP > FRT-myr::smGFP-FLAG in su(Hw)attP1* (*Nern et al., 2015*). All transgenes were in a w1118 background. Behavior experiments used flies with a wildtype X chromosome to permit normal vision.

Experiments were carried out on 3–7 day old heterozygote males. Controls omitted either the *GAL4* driver or the *UAS* effector; for example, for TNT experiments on mn9, the two controls were the mn9 split GAL4 combination with no *UAS-TeTxLC.TNT* ('split control') and *UAS-TeTxLC.TNT* with no split GAL4 ('TNT control'). Flies were raised at 25°C and 50% relative humidity on standard cornmeal and molasses food in a 12 hr light/dark cycle, except for flies for CsChrimson experiments, which were dark-reared on food containing 0.2 mM all-trans retinal (Toronto Research Chemical, #R240000). Behavioral experiments were performed at 25°C, 40% humidity. When needed before experiments, anesthesia was performed on ice.

## Synchrotron

Flies were tethered under cold anesthesia and filmed at 3000 fps with an x-ray beam in the Advanced Photon Source synchrotron at the Argonne National Laboratory. Proboscis extension was filmed during rare instances when it occurred spontaneously.

## CNS immunohistochemistry and imaging

Brains and ventral nervous systems were dissected and stained using published methods (*Nern et al., 2015*). Antibodies: rabbit anti-GFP (1:500, Invitrogen, #A11122), mouse anti-Bruchpilot (1:50, Developmental Studies Hybridoma Bank, University of Iowa, mAb nc82), Alexa Fluor 488-goat anti-rabbit (1:500, ThermoFisher A11034), Alexa Fluor 568-goat anti-mouse (1:500, ThermoFisher A11031). Serial optical sections were obtained at 1 µm intervals on a Zeiss 700 confocal with a Plan-Apochromat 20x/0.8NA objective.

## Preparation of whole mount heads for muscle and cuticle imaging

Flies were anesthetized with $CO_2$ and briefly washed with 70% ethanol. Antennae, maxillary palps, eye cups (including retinas) and a small part of labellum were removed from isolated heads under 2% paraformaldehyde/PBS/0.1% triton X-100. Heads were fixed in this solution overnight at 4°C, then incubated in PBS with 1% triton X-100, 0.5% DMSO and Escin (0.05 mg/ml, Sigma-Aldrich, E1378) containing Texas Red-X Phalloidin (1:50, Life Technologies #T7471) and a chitin-binding dye Calcofluor White (0.1 mg/ml, Sigma-Aldrich #F3543-1G) at room temperature with agitation for up to 5 days. Long incubation times and the presence of surfactants assured better penetration of phalloidin into the muscles. The samples were then washed in PBS/1% triton (4 × 1 hr) and fixed for 4 hr in 2% paraformaldehyde to reduce leaching of bound phalloidin from muscles during the subsequent ethanol dehydration step. To avoid artefacts caused by osmotic shrinkage of soft tissue, samples were gradually dehydrated in glycerol (2–80%) and then ethanol (20% to 100%) (*Ott, 2008*) and mounted in methyl salicylate (Sigma-Aldrich #M6752) for imaging.

## Preparation of head sections for muscle and cuticle imaging

Heads were prepared as described above, washed, embedded in 7% agarose (SeaKem, Lonza #50002) in PBS and sectioned sagittally on a vibratome (Leica VT1000s) at 0.3 mm. The slices were incubated with Texas Red-X Phalloidin and Calcofluor (solution composition as above) for 2 days at room temperature, washed and mounted in Tris-buffered (50 mM, pH 8.4) 50% glycerol. To image cuticle alone, soft tissues were digested away after imaging, with 0.25 mg/ml collagenase/dispase (Roche #10269638001) and 0.25 mg/ml hyaluronidase (Sigma Aldrich #H3884) in PBS for 5 hr at 37°C. After washing, exoskeleton was stained with Congo Red (0.5 mg/mL, Sigma-Aldrich #C676), washed and mounted in buffered glycerol. Serial optical sections were obtained at 1 µm intervals on

a Zeiss 500 or 710 confocal microscopes with a LD-LCI 25x/0.8 NA objective. Congo Red was imaged using a 594 nm laser.

## Whole mount head staining for synapse, muscle and GFP imaging

Type I, II and III boutons all express the presynaptic protein Bruchpilot (*Wagh et al., 2006*), so it was chosen here as a pan-synaptic marker for neuromuscular junctions. Heads were prepared as above, except fixation time was reduced to 2 hr due to the sensitivity of Bruchpilot epitopes to prolonged fixation. The heads were incubated in PBS with aforementioned surfactants and 3% goat serum (Life Technologies #776318), Alexa Fluor 633 Phalloidin (1:50, Life Technologies # A22284), rabbit anti-GFP polyclonal (1:1000, Thermo Fisher, #A10262) and mouse anti-Bruchpilot mouse monoclonal antibodies (1:50 [concentrate]; DSHB/Developmental Studies Hybridoma Bank, mAb nc82) at room temperature with agitation for 5 days. After a series of four ~ 1 hr washes in PBS containing surfactants, the sections were incubated for another 24 hr in the above buffer containing secondary antibodies (1:1000, Alexa Fluor 488 goat anti-rabbit, Thermo Fisher #A32731; 1:250 Cy3 goat anti-mouse, Jackson Immunoresearch #115-166-003). Samples were washed, post-fixed, dehydrated and cleared for imaging as described above. Serial optical sections were obtained at 1 µm intervals on a Zeiss 500 or 710 confocal microscopes with a LD-LCI 25x/0.8 NA objective. Calcofluor White, anti-GFP/anti-rabbit Alexa 488 antibodies, anti-nc82/anti-mouse Cy3 antibodies, Congo Red, Texas Red and Alexa 633 phalloidin-treated samples were imaged using 405, 488, 561, 594 and 633 nm lasers, respectively.

## Stochastic labeling

For multicolor flipout stochastic labeling, flies received a 15 min heat shock at 37°C at 1–3 d old, and were dissected at 6–8 d. Dissections were performed to carefully remove the brain from the head but preserve both for staining, so that an individually labeled neuron in the brain could be correlated with its projections to the muscle. Staining was performed as described (*Nern et al., 2015*), with the following antibodies: rat anti-flag (Novus Biologicals, LLC, Littleton, CO, #NBP1-06712), rabbit anti-HA (Cell Signaling Technology, Danvers, MA, #3724S), mouse anti-V5 (AbD Serotec, Kidlington, England, #MCA1360), AlexaFluor-488 (1:500; goat anti-rabbit, goat anti-chicken, goat anti-mouse; Thermo Fisher Scientific), AlexaFluor-568 (1:500; goat anti-mouse, goat anti-rat; Thermo Fisher Scientific), AlexaFluor-633 (1:500; goat anti-rat; Thermo Fisher Scientific).

## Image processing

Images were adjusted for gain and contrast without obscuring data. Images were processed in ImageJ (https://imagej.nih.gov/ij/), Icy (http://icy.bioimageanalysis.org/) and Photoshop (Adobe Systems Inc). Where noted, motor neurons and muscles were rendered and segmented from confocal stacks with VVDviewer software (https://github.com/takashi310/VVD_Viewer), to visualize them in isolation. For this rendering and for computational alignment of brain images used where noted, brain images were registered using the Computational Morphometry Toolkit (http://nitrc.org/projects/cmtk) (*Jefferis et al., 2007*), to a standard brain template ('JFRC2014') mounted and imaged with the same conditions, which had been corrected by a z scaling factor of 1.568 to match the true proportions of a fly brain (obtained from frontal vs. horizontal stacks of the brain [*Ito et al., 2014*]).

## Occupancy experiments

Head staining (described above) was used to determine whether the collection included every primary motor neuron of the proboscis. To be sure that every axon terminal was accounted for, we stained each motor neuron line one by one, and inspected every bouton in the 3D stack. All the boutons on the relevant muscle were indeed completely occupied by the labeled motor neurons. The lack of unoccupied boutons suggests that modulatory motor neurons' boutons in the proboscis may not be labeled by the synaptic marker used here, or may not be visible as separate punctae since modulatory boutons can lie directly on top of primary motor neuron terminals (*Rivlin et al., 2004*). Incomplete occupancy of the boutons of muscle 3 was to be expected, since no line could be found to cover all its motor neurons, so separate lines were used that targeted muscle 3's medial and lateral fibers. For muscle 10, a bilateral motor neuron pair was never observed in 6 stained heads, only a single unilateral neuron that could be found in either hemisphere. Since it did not occupy all

boutons on muscle 10, we infer that it is not an unpaired neuron, but stochastic expression in a pair of motor neurons.

## Cameras and lighting

Behavior was filmed with Basler A622f cameras at 30 Hz, controlled by gVision software developed at Janelia (Gus Lott). Ambient lighting was provided by white light, or by two infrared security spotlights (Phenas) for CsChrimson experiments. For CsChrimson activation, a 656 nm LED spotlight (Mightex PLS-0656–101 C) was controlled through a NIDAQ board and Labview software (National Instruments, Austin, TX). An 880 nm infrared LED (Fairchild Semiconductor Corporation #QEE123) was connected to a fiber optic in the field of view, to indicate when the stimulus was on, since a long-pass 830 nm filter in front of the camera prevented the stimulus light from obscuring the behavior video. Each stimulus was 2 s of constant light at 100 µW/mm$^2$.

## Freely moving behavior

To compare proboscis extension during feeding vs. courtship, flies were placed in oval chambers, 3 mm high, 9 mm wide and 12 mm long, to provide a flat segment along the wall in which to film proboscis extension from the side. For feeding, wildtype males were starved with access to water for 24 hr, then loaded into chambers with 10% sucrose/2% agarose painted onto a segment of the wall. Proboscis angles were scored in the frame of greatest proboscis extension during one proboscis extension event (event = time from extension to retraction), in events where the fly was orthogonal to the camera, contacting the sucrose on the wall. Courtship was filmed in chambers of the same dimensions but with a virgin female present. Proboscis extension was likewise scored in the frame of maximum extension, in courtship licking events occurring when the male was orthogonal to the camera and directly behind the female on the flat segment of wall.

## Tethered fly behavior

Flies were anesthetized using cold, and mounted by the thorax to a pin with UV-cured glue (LED-200 UV gun, Electro-Lite Corp.). The same glue was used in experiments in which proboscis segments were glued. Mounted flies were allowed to recover for a few minutes before CsChrimson stimulation experiments, or alternatively were deprived of food and water for 5 hr before sucrose presentation experiments.

For sucrose presentation, a small 100 mM sucrose droplet was presented to the legs from a syringe on a manipulator, holding the target at a distance that prevents proboscis contact. The proboscis is extended repeatedly in search of the sucrose, moving through a range of angles (*Video 7*). The legs are not in contact with the ground and wave almost continuously, therefore rapidly encounter the sucrose. The proboscis reaching is guided not by vision but by leg contact, since the assay is performed under infrared illumination not visible to the fly. Visually-guided proboscis reaching was never observed in these conditions; the proboscis never extended before the legs tasted the sucrose.

## Proboscis, head and target angle measurements

Joint angles for the head, rostrum and haustellum were measured from individual video frames in ImageJ by marking the points shown in *Figure 2C–D*. The rostrum angle is closer to 180˚ than 0˚ at rest, so is expressed as subtracted from 180˚ so that values are smaller at rest than during proboscis extension. When the haustellum is drawn into the head, its angle cannot be directly measured, but can be extrapolated by trigonometry, using the law of cosines with the distance from the rostrum joint to the end of the haustellum, and assuming that the rostrum and haustellum lengths are fixed proportions of the measured size of the head.

Measures of joint angles were chosen to best suit each assay. Due to the challenge of gluing flies to pins manually, the camera view varied, leading to variability in the measured raw angles of the joints. To overcome this variability, within-fly normalization was performed wherever possible, subtracting food- or CsChrimson-stimulated extension angle from the angle immediately before stimulation. Note that normalization was not possible for resting state angles, since there is no change from baseline. Analysis of raw joint angles (*Figure 7—figure supplement 2*) shows similar relations between motor neurons and joint angles as the normalized data but with greater inter-fly variability,

as expected from the range of starting positions, so the main figures use normalized values as follows. Wildtype joint angles (*Figure 2M–O*) were measured in the frame 200 ms after the beginning of proboscis movement elicited by sucrose. For CsChrimson stimulation from rest, angles were measured in the frame of maximal proboscis movement during the stimulus and subtracted from the frame before the stimulus to report change in angle. For CsChrimson stimulation during ongoing feeding, angles were measured in the frame 200 ms after the stimulus start and subtracted from the frame before the stimulus. For TNT silencing, angles were measured at rest (where noted in figures), or during feeding (as the difference in angle from the start of proboscis extension to 200 ms later). During CsChrimson activation from rest, proboscis change is total extension of proboscis (farthest point on back of head to tip of labellum) at maximum movement divided by at rest (frame before stimulus).

During TNT silencing, proboscis position provides a measure of how retracted the proboscis is at rest, and is measured as the distance from back of head to tip of labellum divided by this value averaged across controls, after first normalizing the size of the fly (using measured height of head). Labellar extension and abduction were quantified as binary values, occurring or not occurring.

The angle of the sucrose target was measured from a line along the posterior-anterior axis of the fly (drawn from a point on the thorax marked by the humeral bristles, to the center of the eye), in the first frame of leg contact with the sucrose. A target at 90°=directly ventral to the eye.

Mean reach angle was measured by creating a stack in ImageJ (https://imagej.nih.gov/ij/) of the trial video frames, and generating an average intensity projection, creating a cloud of pixels where the proboscis had moved. In the region where the proboscis tip had moved, the pixel of maximum brightness was taken as the mean location of the tip over the trial. Mean reach angle was calculated from this proboscis tip location, with respect to a line along the posterior-anterior axis of the fly (drawn from the humeral bristles to the center of the eye).

To determine the accuracy of reaching, aim deviation was calculated as mean reach angle minus target angle, so that a deviation above 0° would represent reaching above the target in the dorsoventral plane, and aim below 0° would be below. Physical constraints may affect aim; flies appear to reach too high to low targets, and too low to high targets (*Figure 2L*). For proboscis reaching in wildtype flies (*Figure 2*), occasional flies were excluded if they did not appear hungry after the deprivation period (no proboscis extension to sucrose).

## Speed, velocity, length and distance measurements

Points were manually drawn on video frames in ImageJ to acquire coordinates, which were converted to lengths by basic trigonometry. Frame 1 was taken to be the frame of the first rapid acceleration of the proboscis after sucrose contact. Target location was taken to be site of leg contact with sucrose in most recent frame of contact before proboscis extension. Reach speed = distance proboscis tip traveled per ms (involving both proboscis and/or head motion, in any direction). Proboscis velocity = tip displacement towards target minus head displacement towards target, per ms. Proboscis length = distance from rostrum joint to tip, subtracting mean length at start to set starting length to zero. Proboscis distance from target = distance from tip to target.

## Statistical information

Experiments were blinded for scoring by assigning random identifications and only unblinding after all scoring. T-tests were two-tailed. Kolmogorov–Smirnov tests were first used to confirm that the data were consistent with a normal distribution. Error bars are standard error of the mean (SEM). In jitter plots, horizontal bars represent the mean. Outliers were not excluded. Sample size and other statistical information is provided in each relevant figure legend or Methods section. Asterisks: *$p < 0.05$, **$p < 0.01$, ***$p < 0.001$, N.S.: not significant. All replicates are biological, testing different flies, not retesting the same individuals as technical replicates. Benjamini-Hochberg multiple testing correction used a false discovery rate of 0.1, and was applied across every genotype tested (across all motor neuron testing figures). In comparing mean reach angle to high and low targets, an a priori power analysis was performed in G*Power software using an effect size of 0.7 (estimated from a pilot study), an alpha error probability of 0.05, one tail, and a power of 0.8, giving a required sample size of 28 per group.

## Acknowledgements

For fly strains, we thank Barret Pfeiffer, Gerry Rubin, and Vivek Jayaraman. For technical support, we thank Phuong Chung and the Janelia Fly Core, Instrument Design & Fabrication, Project Technical Resources, and Scientific Computing groups (particularly Hideo Otsuna). For synchrotron movies, we thank John Tuthill and Nikolay Kladt. Use of the Advanced Photon Source was supported by the US Department of Energy, Office of Science, Office of Basic Energy Sciences, under Contract No. DE-AC02-06CH11357, with technical support from Wah-Keat Lee. This work was made possible in part by software funded by the NIH: Fluorender: An Imaging Tool for Visualization and Analysis of Confocal Data as Applied to Zebrafish Research, R01-GM098151-01.

## Additional information

### Funding

| Funder | Grant reference number | Author |
|---|---|---|
| Howard Hughes Medical Institute | Janelia Research Campus | Claire E McKellar<br>Igor Siwanowicz<br>Barry Dickson<br>Julie H Simpson |

The funders had no role in study design, data collection and interpretation, or the decision to submit the work for publication.

### Author contributions

Claire E McKellar, Conceptualization, Data curation, Formal analysis, Validation, Visualization, Methodology, Writing - original draft, Project administration; Igor Siwanowicz, Investigation, Methodology; Barry J Dickson, Resources, Supervision, Funding acquisition, Project administration, Writing - review and editing; Julie H Simpson, Conceptualization, Resources, Supervision, Funding acquisition, Project administration, Writing - review and editing

### Author ORCIDs

Claire E McKellar  https://orcid.org/0000-0002-3580-7336
Igor Siwanowicz  http://orcid.org/0000-0001-5819-1530
Barry J Dickson  https://orcid.org/0000-0003-0715-892X
Julie H Simpson  https://orcid.org/0000-0002-6793-7100

### Decision letter and Author response

Decision letter https://doi.org/10.7554/eLife.54978.sa1
Author response https://doi.org/10.7554/eLife.54978.sa2

## Additional files

### Supplementary files

• Transparent reporting form

### Data availability

All data generated or analysed during this study are included in the manuscript and supporting files.

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
