## [Decision Letter]

Thank you for submitting your article "Comprehensive genetic tools to target motor neurons for each muscle of the fly proboscis" for consideration by *eLife*. Your article has been reviewed by three peer reviewers, including Ronald Calabrese as the Reviewing Editor, Senior Editor and Reviewer #2.

The reviewers have discussed the reviews with one another and the Reviewing Editor has drafted this decision to help you prepare a revised submission.

Summary:

This is a co-submission of two papers. The goal of the first work is achieving genetic control over motor neurons for every muscle of the *Drosophila* proboscis, and it is accomplished convincingly. The degree of success constructing split Gal4 lines that label only a single bilateral pair of motor neuron corresponding to each muscle of the proboscis is really remarkable, and there is no doubt that these tools will be of tremendous use in defining the neuronal networks responsible for reaching movements of the proboscis and indeed the sucking movements also. The anatomical data presented is beautiful. After achieving genetic control over motor neurons for every muscle of the *Drosophila* proboscis, the authors in the second work systematically test every motor neuron that inserts on cuticle (including apodemes) and is thus likely to move the proboscis (rather than inserting on the pharynx and likely involved in sucking) with activation and inactivation experiments. By combining kinematics and some biomechanics, the authors arrive at a functional role for each of the proboscis inserting muscle and their motor neurons. The data presented is comprehensive and convincing. When combined these two papers, very clearly supersede a previous attempt (Schwarz et al., 2017) at such tool development and analysis with several important corrections and many major refinements.

Essential revisions:

Upon careful review of these two co-submitted papers (henceforth referred to a Paper 1 and Paper 2) it was the clear consensus of the reviewers that neither paper alone is strong enough for *eLife* and we would understand if you chose to take the paper(s) elsewhere. If on the other hand, you chose to submit the two papers to *eLife* as one joint paper, then we would ensure that they were sent to the same reviewers, and we think that there is a high chance of publication with a short turn-around. No new experiments or analyses are needed.

If you take the *eLife* route the papers should be combined and focused. One paper will be easier for the authors to present and for the reader to comprehend. One could not read the Paper 2 without constantly referring to Paper 1 and the Paper 2 contains much repetition of the findings of Paper 1.

There are several ways that the papers can be fitted together as detailed in the individual expert reviews.

One potential path:

1) The theme presented for Paper 1 of rhythmic vs. discrete behaviors is not effectively exploited nor can it be given the data presented. Likewise, the comparison to human reaching in Paper 2 seems premature and focusing the combined single paper on the tools developed and the wonderful anatomy and first-pass functional analysis seems more fruitful. The data Figure 9 Paper 1 seems premature and underdeveloped for publication as does the data of Figure 8 Paper 2. They are presented is support of the rhythmic vs. discrete behaviors and human reaching themes and are not compelling. We recommend removing these figures to focus the paper.

2) Efficiency with the figures would help. Figure 1 of Paper 2 could benefit from comparison directly to Figure 1 of Paper 1. Figure 2 of Paper 1 should be supplementary and fits thematically with Paper 2 material. The expert reviews suggest other figure fusions and changes.

3) The data of Figure 7 Paper 1 is reminiscent of data in Schwarz et al., 2017, albeit with corrections but need not be elaborated and could be perhaps combined with Figure 8 Paper 1, which is somewhat underdeveloped.

4) Discussions can be shortened and in Paper 2 shifted away from detailed comparison to in Schwarz et al., 2017.

5) All Essential revisions in the expert reviews should all be addressed.

Reviewer #1:

In two related papers, McKellar and colleagues present the results of a screen to genetically label and optogenetically activate each motor neuron that innervates muscles in the proboscis of *Drosophila melanogaster*.

In the first manuscript, entitled "Comprehensive genetic tools to target motor neurons for each muscle of the fly proboscis", they present genetic driver lines for each motor neuron together with careful anatomy of cuticular structures and musculature. Ten proboscis muscles and their associated motor neurons control the position of proboscis parts and are involved in the proboscis extension response (PER), while the remaining eight muscles control the constriction of the pharynx. In the course of this work, the authors identify several new muscles.

Once the authors develop split-Gal4 genetic labels for each motor neuron, they can drive the expression of CsChrimson to activate, or Tetanus toxin to silence, each motor neuron. They present the results of these experiments in a second paper entitled "Motor neuron control of *Drosophila* proboscis reaching". The main advances of this work are the resulting genetic tools and a more comprehensive knowledge of how particular muscles insert onto cuticular structures to extend or flex proboscis parts about joints. These advances clarify the results of past investigations similar into motor control of the proboscis (in particular, Schwarz, Motor control of *Drosophila* feeding behavior).

Across both papers, the images of motor neuron and musculature anatomy are illustrative and quite beautiful. Optogenetic activation of individual motor neurons confirms the role each plays in controlling the proboscis segment. Together, these data allow the authors to articulate a model of how motor neuron activation and muscle contraction enables the fly to extend its proboscis.

Although the tools and data in these papers are detailed and valuable, the papers do not effectively show what new insight has been gained relative to previous studies. There is also a large experimental gap: while the authors create the necessary genetic tools, they do not investigate the neural control and natural dynamics of proboscis movement beyond artificial optogenetic stimulation. In the absence of such data, the main contribution of these papers is the clear and thorough anatomy. However, the convoluted organization of the two manuscripts require the reader to sift through disparate data to reach a clear understanding of how the proboscis motor system is organized. These manuscripts would need to be significantly reorganized and streamlined to be understood by a general neuroscience audience.

Essential revisions:

1) It is not clear to why these manuscripts are separate. The basis for separating them seems to be to avoid one behemoth paper but the division between tools and function obscures the take-home message of the work. For example, many of the descriptions of musculature in "Comprehensive genetic tools to target motor neurons for each muscle of the fly proboscis" (called paper 1 below) are critical for understanding the optogenetic activation of motor neurons in "Motor neuron control of *Drosophila* proboscis reaching" (called paper 2 below), resulting in an unnecessary duplication of effort between the manuscripts to understand how the proboscis is moved. Take mn9, for example: it is hard to know what to take from Figure 1, Figure 4 and Figure 9 in paper 1, if the critical proof to show that mn9 extends the proboscis is in paper 2. (Past work has shown m9 to be involved in proboscis protraction, so you could argue that a knowledgeable reader may know this, but that undercuts the purpose of Figure 2 in paper 2). There are also examples of data in primary figures that do not advance the author's case. For example, in paper 1, Figure 2, Figure 3, Figure 6 and Figure 8 are supplementary figures. Figure 2: The images of FlyLight Gen 1 Gal4 lines are unhelpful, fly researchers can find these and they are not helpful to a non-Drosophilist. Figure 3: I commend the authors for including images of the entire nervous system, not often shown, but again, these are supporting information. Figure 6 can support Figure 5. Figure 8 does suggest that MNs likely do not receive direct information from sensory neurons or other previously described interneurons, but these hypotheses are not tested further, so Figure 8 can support Figure 7.

By the same token, paper 2 would benefit from schematics of muscles and an overview of reaching related vs. pharyngeal motor neurons. This distinction is helpful, and since it is consistent with past work. Because the function of pharyngeal muscles is not explored further, it would not detract from the message of paper 2. paper 2 contains functional data that can be combined: e.g. protraction (Figure 2) and retraction (Figure 3) of the rostrum could be one figure, together with anatomical data from paper 1. Figure 2, Figure 3, Figure 4 and Figure 5 also contain many panels of different metrics, normalized or baseline subtracted. It is not necessary to show all these data in this form, and it detracts from the message. My suggestion would be to combine Figure 2 Figure 3, Figure 4 and Figure 5 haustellum and rostrum images. Figure 7 is beautiful! But it should be split up to accompany the data in Figure 2 and Figure 3 and Figure 4 and Figure 5.

In addition, much of the introductions of both manuscripts are either unnecessary or misleading (see below), so combining the data into a single manuscript will also help to eliminate those problems. In summary, I do not see a clear reason to split data across papers.

2) The optogenetic activation data that the authors present do not provide insight into how reaching kinematics are produced: 2 second stimulation of motor neurons is not a naturalistic manipulation, and resulting proboscis trajectories are collapsed into single numbers in Figure 2, Figure 3, Figure 4, Figure 5 and Figure 6 of paper 2. Schwarz et al., also concluded that reaching is the result of precise temporal patterns of activation of proboscis motor neurons. The authors have all the tools to explore this further with calcium imaging or electrophysiology, though it is understandably beyond the scope of this paper. I anticipate another paper that begins with the observations in Figure 8 of paper 2 and then uses the tools shown here to explore how these searching movements are created.

3) A related aspect of reaching is the treatment of what the authors call "coupling" of distinct parts of the proboscis, as when mn9 is activated and extends the haustellum in addition to the rostrum. Is this coupling mechanical, neural, or driven by proprioceptive feedback? The activation of additional motor neurons could be ruled out by silencing haustellum extending mns while activating mn9. The authors do not discuss sensory feedback, or the role of proprioception in determining the location of the target. This would be an interesting direction to pursue to draw closer connections to reaching in other organisms.

4) The authors state at various times that these genetic tools could help to make neural control of the proboscis a model for (1) rhythmic vs. discrete movements, and (2) a model of reaching in humans. These statements appear in the abstract and introductions of both papers, with little scholarship or links to literature that may prime the reader as to functional features that are similar in both models. The data to support these assertions are that proboscis speed is a bell shaped curve as a function of time and that proboscis reaching pauses to enter a second phase of slow approach. These observations seem rather trivial; what is the alternative? As the authors point out, proboscis reaching is already thought to be more complex than simply a reflex.

Reviewer #2:

Achieving genetic control over motor neurons for every muscle of the *Drosophila* proboscis is the laudable goal of this work, and it is accomplished convincingly. Surely there are nits to pick over the failure the achieve split Gal4 lines that label only a single bilateral pair of motor neuron corresponding to each muscle of the proboscis, BUT the degree of success is really remarkable, and there is no doubt that these tools will be of tremendous use in defining the neuronal networks responsible for reaching movements of the proboscis and indeed the sucking movements also. The anatomical data presented is beautiful. When combined with the companion paper (Motor neuron control of *Drosophila* proboscis reaching) it is very clear that this work supersedes a previous attempt (Schwarz et al., 2017) at such tool development and analysis with several important corrections and many major refinements

Essential revisions:

Upon carefully reviewing this paper (henceforth referred to a Paper 1) and its companion by the exact same authors (Motor neuron control of *Drosophila* proboscis reaching – henceforth referred to as Paper 2), I do not recommend that they be published independently. One paper will be easier for the authors to present and for the reader to comprehend. I could not read the Paper 2 without constantly referring to Paper 1 and the Paper 2 contains much repetition of the findings of Paper 1. The theme presented for Paper 1 of rhythmic vs. discrete behaviors is not effectively exploited nor can it be given the data presented so much of the rationale for an independent paper is lost. There are several ways that the papers can be fitted together. Efficiency with the figures would help. Figure 1 of Paper 2 could benefit from comparison directly to Figure 1 of Paper 1. Figure 2 of Paper 1 should be supplementary and fits thematically with Paper 2. The VNC confocal projection of Figure 3 Paper 1 are distracting, convey little information useful to the general reader, and can certainly be supplementary. The data of Figure 7 Paper 1 is reminiscent of data in Schwarz et al., 2017, albeit with corrections but need not be elaborated and could be perhaps combined with Figure 8 Paper 1 which is somewhat underdeveloped. The data Figure 9 Paper 1 seems premature for publication.

Reviewer #3:

This work provides tools that target every type of motor neurons in the proboscis, as evidenced by the fact that all muscle types are covered by the domains of neuromuscular junctions. Proximity of motor neuron dendrites to other neurons provide hints as to the interaction of sensory and central circuits with the motor neurons. These tools can now be used for functional studies, an example of which is provided in the accompanying manuscript. I found this to be really nice manuscript that will certainly be very useful for the *Drosophila* neural circuit community, particularly those working on feeding behavior. It really fills a missing dataset, complementing the numerous sensory neuron lines that are available, as well as the increasing number of interneuronal lines that have been used for feeding behavior analysis. My comments are not of the major kind, focusing more on presentation style.

[Editors' note: further revisions were suggested prior to acceptance, as described below.]

Thank you for resubmitting your article "Controlling every muscle for directed reaching of the fly proboscis" for consideration by *eLife*. Your article has been reviewed by three peer reviewers, including Ronald Calabrese as the Reviewing Editor, Senior Editor and Reviewer #2.

The reviewers have discussed the reviews with one another and the Reviewing Editor has drafted this decision to help you prepare a revised submission.

Summary:

This paper presents remarkable anatomy and structure-function analysis of the muscles controlling the proboscis of *Drosophila* and firmly establishing this as model for the study of directed movements. They also perform thorough motor neuron activation (CsChrimson) /silencing (TNT) experiments using highly specific splitGAL4 lines and the lines provide a great set of tools for deeper functional analysis going forward.

Title: Please consider the changes to the title suggested by reviewer #1.

Essential revisions:

The authors have made a very strong revision of the manuscript, responding appropriately to all the reviewer comments. In the spirit of make a strong paper even stronger, the reviewers present further revisions, that can be accomplished expeditiously and can be reviewed by the RE.

Reviewer #1:

Summary:

This revised manuscript, in which the authors combined their two previous manuscripts, is much improved. I have one point that should be addressed to help improve the clarity and organization of the paper.

Essential revisions:

Figure 7 and Figure 8 contain many separate panels that describe the effects of activation and deficits of silencing proboscis motor neurons. Even though the y-axis labels contain the information they report, each panel uses different units, which makes it confusing for readers to understand how these values relate to proboscis kinematics.

In particular, interpreting changes in angle during feeding requires extra mental gymnastics. The change in angle from rest to feeding appears to be 80 deg (Figure 7I and Figure 7J). Activating mn9 does not further increase the rostrum feeding angle in Figure 7D. One might then attempt to compare angles in Figure 7I and Figure 7D, but the authors are thinking about this differently. Their logic is that activation (Figure7D) happens after the proboscis is extended, so the change should be about 0, whereas the positive 80 deg change during feeding does not occur when mn9 is silenced.

I suggest that instead of displaying the change in rostrum and haustellum angle during feeding, the authors should plot rostrum and haustellum angle during feeding. The authors should also explain the rationale behind the units in the legend. It is possible that chrimson expression in mn9 actually causes a different feeding angle to begin with, which would be interesting. If the proboscis was more extended, this would suggest an increase in baseline excitation. If it was more flexed and retracted, it could imply compensatory feedback to combat overexcitation. Same for mns 1 and 2V.

Reviewer #3:

What a gorgeous paper. I think it's a wonderful piece of work. I only have two very minor comments on the revision:

1) Subsection “An updated inventory of proboscis muscles” (second paragraph). Either a semi-colon or full stop after m3M. Also, in the figure legend, define L at the bottom.

2) In the Abstract, they write "…and throughout entire circuits, as the fly connectome approaches." However, there is no mention of this theme in the Discussion section. Maybe a sentence or two somewhere in the Discussion section on how their identification of the muscles and the motor neurons might help decipher the meaning of the interneuronal connections that will be revealed in the connectome. I personally believe it will be enormously useful, because, as the authors write (Introduction), "…they are the outputs upon which higher order circuits must act."

---

## [Author Response]

Essential revisions:Upon careful review of these two co-submitted papers (henceforth referred to a Paper 1 and Paper 2) it was the clear consensus of the reviewers that neither paper alone is strong enough for eLife and we would understand if you chose to take the paper(s) elsewhere. If on the other hand, you chose to submit the two papers to eLife as one joint paper, then we would ensure that they were sent to the same reviewers, and we think that there is a high chance of publication with a short turn-around. No new experiments or analyses are needed.If you take the eLife route the papers should be combined and focused. One paper will be easier for the authors to present and for the reader to comprehend. One could not read the Paper 2 without constantly referring to Paper 1 and the Paper 2 contains much repetition of the findings of Paper 1.There are several ways that the papers can be fitted together as detailed in the individual expert reviews.One potential path:1) The theme presented for Paper 1 of rhythmic vs. discrete behaviors is not effectively exploited nor can it be given the data presented.

Now removed.

Likewise, the comparison to human reaching in Paper 2 seems premature and focusing the combined single paper on the tools developed and the wonderful anatomy and first-pass functional analysis seems more fruitful.

We understand the reservations about the analogy to directed reaching in vertebrates, and have removed the more speculative mention of this idea. However, since we do believe this to be an exciting future application of this work, we include it briefly in the Discussion section.

The data Figure 9 Paper 1 seems premature and underdeveloped for publication as does the data of Figure 8 Paper 2. They are presented is support of the rhythmic vs. discrete behaviors and human reaching themes and are not compelling. We recommend removing these figures to focus the paper.

Done.

2) Efficiency with the figures would help. Figure 1 of Paper 2 could benefit from comparison directly to Figure 1 of Paper 1.

Now reorganised.

Figure 2 of Paper 1 should be supplementary and fits thematically with Paper 2 material.

now removed for the sake of streamlining, since it made only a minor point.

The expert reviews suggest other figure fusions and changes.

Done.

3) The data of Figure 7 Paper 1 is reminiscent of data in Schwarz et al., 2017, albeit with corrections but need not be elaborated and could be perhaps combined with Figure 8 Paper 1, which is somewhat underdeveloped.

We have handled this according to Reviewer 1’s request that the former Figure 8 be a supplement to the former Figure 7 – now Figure 5.

4) Discussions can be shortened and in Paper 2 shifted away from detailed comparison to in Schwarz et al., 2017.

We have removed the detailed comparison from the text. There are differences in the findings that significantly alter interpretation of muscle roles, so wenow provide a supplementary table (newFigure 9 —figure supplement 1) summarizing our findings, and showing with asterisks which are new or different from published work.]

5) All Essential revisions in the expert reviews should all be addressed.

Done.

Reviewer #1:Although the tools and data in these papers are detailed and valuable, the papers do not effectively show what new insight has been gained relative to previous studies.

Please see newFigure 9 —figure supplement 1 summarizing that 12 of the 16 behavioral phenotypes we describe were previously unknown.

There is also a large experimental gap: while the authors create the necessary genetic tools, they do not investigate the neural control and natural dynamics of proboscis movement beyond artificial optogenetic stimulation.

While optogenetic activation is artificial, silencing experiments are carried out in flies performing natural behaviors (for example, in Figure 8F-J and Figure 9H-I). Controls in these experiments provide measures of natural behavior. Besides these experiments, natural behavior was also tested in wildtype flies using five other measures (Figure 2E-G, K-L), plus an additional four now removed by reviewer request (previous paper 2, Figure 8). Certainly, more characterization of natural behavior will be needed in the future to describe precise kinematics.

In the absence of such data, the main contribution of these papers is the clear and thorough anatomy. However, the convoluted organization of the two manuscripts require the reader to sift through disparate data to reach a clear understanding of how the proboscis motor system is organized. These manuscripts would need to be significantly reorganized and streamlined to be understood by a general neuroscience audience.

Now combined into one paper.

Essential revisions:1) It is not clear to why these manuscripts are separate. The basis for separating them seems to be to avoid one behemoth paper but the division between tools and function obscures the take-home message of the work. For example, many of the descriptions of musculature in "Comprehensive genetic tools to target motor neurons for each muscle of the fly proboscis" (called paper 1 below) are critical for understanding the optogenetic activation of motor neurons in "Motor neuron control of *Drosophila* proboscis reaching" (called paper 2 below), resulting in an unnecessary duplication of effort between the manuscripts to understand how the proboscis is moved. Take mn9, for example: it is hard to know what to take from Figure 1, Figure 4 and Figure 9 in paper 1, if the critical proof to show that mn9 extends the proboscis is in paper 2. (Past work has shown m9 to be involved in proboscis protraction, so you could argue that a knowledgeable reader may know this, but that undercuts the purpose of Figure 2 in paper 2). There are also examples of data in primary figures that do not advance the author's case. For example, in paper 1, Figure 2, Figure 3, Figure 6 and Figure 8 are supplementary figures.

Now reorganized.

Figure 2: The images of FlyLight Gen 1 Gal4 lines are unhelpful, fly researchers can find these and they are not helpful to a non-Drosophilist.

now removed.

Figure 3: I commend the authors for including images of the entire nervous system, not often shown, but again, these are supporting information.

Done.

Figure 6 can support Figure 5.

We have made room in other ways so that this can remain a main figure, since it is a key theme of the paper that insertion sites predict muscle function.

Figure 8 does suggest that MNs likely do not receive direct information from sensory neurons or other previously described interneurons, but these hypotheses are not tested further, so Figure 8 can support Figure 7.

Done.

By the same token, paper 2 would benefit from schematics of muscles and an overview of reaching related vs. pharyngeal motor neurons. This distinction is helpful, and since it is consistent with past work. Because the function of pharyngeal muscles is not explored further, it would not detract from the message of paper 2. paper 2 contains functional data that can be combined: e.g. protraction (Figure 2) and retraction (Figure 3) of the rostrum could be one figure,

now combined

together with anatomical data from paper 1.

Done.

Figure 2, Figure 3, Figure 4 and Figure 5 also contain many panels of different metrics, normalized or baseline subtracted. It is not necessary to show all these data in this form, and it detracts from the message.

Each panel shows a different behavior without redundancy; all are necessary for a full description of each motor neuron’s phenotype. We now streamline by showing extensors and retractors in the same plots.

My suggestion would be to combine Figure 2 Figure 3, Figure 4 and Figure 5 haustellum and rostrum images.

Previous 2-4 and 2-5 now combined.

Figure 7 is beautiful! But it should be split up to accompany the data in Figure 2 and Figure 3 and Figure 4 and Figure 5.

Great suggestion – done.

In addition, much of the introductions of both manuscripts are either unnecessary or misleading (see below), so combining the data into a single manuscript will also help to eliminate those problems. In summary, I do not see a clear reason to split data across papers.

Due to another request, we have removed the former Paper 2, Figure 8, but this had shown an un-collapsed quantification of how proboscis speed, velocity, length and distance varied over time.

2) The optogenetic activation data that the authors present do not provide insight into how reaching kinematics are produced: 2 second stimulation of motor neurons is not a naturalistic manipulation, and resulting proboscis trajectories are collapsed into single numbers in Figure 2, Figure 3, Figure 4, Figure 5 and Figure 6 of paper 2.

We agree.

Schwarz et al., also concluded that reaching is the result of precise temporal patterns of activation of proboscis motor neurons. The authors have all the tools to explore this further with calcium imaging or electrophysiology, though it is understandably beyond the scope of this paper. I anticipate another paper that begins with the observations in Figure 8 of paper 2 and then uses the tools shown here to explore how these searching movements are created.

We agree with this assessment as a future direction, and mention this need in the Discussion.

3) A related aspect of reaching is the treatment of what the authors call "coupling" of distinct parts of the proboscis, as when mn9 is activated and extends the haustellum in addition to the rostrum. Is this coupling mechanical, neural, or driven by proprioceptive feedback? The activation of additional motor neurons could be ruled out by silencing haustellum extending mns while activating mn9. The authors do not discuss sensory feedback, or the role of proprioception in determining the location of the target. This would be an interesting direction to pursue to draw closer connections to reaching in other organisms.4) The authors state at various times that these genetic tools could help to make neural control of the proboscis a model for (1) rhythmic vs. discrete movements, and (2) a model of reaching in humans. These statements appear in the abstract and introductions of both papers, with little scholarship or links to literature that may prime the reader as to functional features that are similar in both models. The data to support these assertions are that proboscis speed is a bell shaped curve as a function of time and that proboscis reaching pauses to enter a second phase of slow approach. These observations seem rather trivial; what is the alternative?

Figure now removed, as per editor’s suggestion, and topics now moved to Discussion as future areas of research.

As the authors point out, proboscis reaching is already thought to be more complex than simply a reflex.Reviewer #2:Essential revisions:Upon carefully reviewing this paper (henceforth referred to a Paper 1) and its companion by the exact same authors (Motor neuron control of *Drosophila* proboscis reaching – henceforth referred to as Paper 2), I do not recommend that they be published independently. One paper will be easier for the authors to present and for the reader to comprehend. I could not read the Paper 2 without constantly referring to Paper 1 and the Paper 2 contains much repetition of the findings of Paper 1. The theme presented for Paper 1 of rhythmic vs. discrete behaviors is not effectively exploited nor can it be given the data presented so much of the rationale for an independent paper is lost.

Theme now removed.

There are several ways that the papers can be fitted together. Efficiency with the figures would help. Figure 1 of Paper 2 could benefit from comparison directly to Figure 1 of Paper 1.

Now reorganised.

Figure 2 of Paper 1 should be supplementary and fits thematically with Paper 2.The VNC confocal projection of Figure 3 Paper 1 are distracting, convey little information useful to the general reader, and can certainly be supplementary.

Done.

The data of Figure 7 Paper 1 is reminiscent of data in Schwarz et al., 2017, albeit with corrections but need not be elaborated and could be perhaps combined with Figure 8 Paper 1 which is somewhat underdeveloped.

We have handled this according to Reviewer 1’s request that the former Figure 8 be a supplement to the former 7, now Figure 5.

The data Figure 9 Paper 1 seems premature for publication.

Now removed.[Editors' note: further revisions were suggested prior to acceptance, as described below.]

Title: Please consider the changes to the title suggested by reviewer #1.Essential revisions:The authors have made a very strong revision of the manuscript, responding appropriately to all the reviewer comments. In the spirit of make a strong paper even stronger, the reviewers present further revisions, that can be accomplished expeditiously and can be reviewed by the RE.Reviewer #1:Essential revisions:Figure 7 and Figure 8 contain many separate panels that describe the effects of activation and deficits of silencing proboscis motor neurons. Even though the y-axis labels contain the information they report, each panel uses different units, which makes it confusing for readers to understand how these values relate to proboscis kinematics.In particular, interpreting changes in angle during feeding requires extra mental gymnastics. The change in angle from rest to feeding appears to be 80 deg (Figure 7I and Figure 7J). Activating mn9 does not further increase the rostrum feeding angle in Figure 7D. One might then attempt to compare angles in Figure 7I and Figure 7D, but the authors are thinking about this differently. Their logic is that activation (Figure7D) happens after the proboscis is extended, so the change should be about 0, whereas the positive 80 deg change during feeding does not occur when mn9 is silenced.I suggest that instead of displaying the change in rostrum and haustellum angle during feeding, the authors should plot rostrum and haustellum angle during feeding. The authors should also explain the rationale behind the units in the legend.

The units for Figure 7 are briefly explained in the legend, which also now directs readers to more information in the Materials and methods section.

It is possible that chrimson expression in mn9 actually causes a different feeding angle to begin with, which would be interesting. If the proboscis was more extended, this would suggest an increase in baseline excitation. If it was more flexed and retracted, it could imply compensatory feedback to combat overexcitation. Same for mns 1 and 2V.

Figure 7 shows how motor neuron activations change proboscis angle. We chose normalized angles because the way we mount flies results in different starting positions, which makes raw (absolute) angles more variable. We expand the explanation for using normalization in Materials and methods section, and direct the reader there from both the text and the legend of Figure 7. We also include a new Figure 7 – supplement with an analysis of the raw angles, as the reviewer requested.

The expanded explanation reads, “Measures of joint angles were chosen to best suit each assay. Due to the challenge of gluing flies to pins manually, the camera view varied, leading to variability in the measured raw angles of the joints. To overcome this variability, within-fly normalization was performed wherever possible, subtracting food- or CsChrimson-stimulated extension angle from the angle immediately before stimulation. Note that normalization was not possible for resting state angles, since there is no change from baseline. Analysis of raw joint angles (Figure 7—figure supplement 2) shows similar relations between motor neurons and joint angles as the normalized data but with greater inter-fly variability, as expected from the range of starting positions, so the main figures use normalized values as follows…”

The relevant data panels for comparison of normalized vs. raw are:

Normalized: Raw:

Figure 7D, Figure 8D, Figure 9M Figure 7—figure supplement 2A

Figure 7E, Figure 8E, Figure 9N Figure 7—figure supplement 2B

Figure 7I, Figure 8I, Figure 9R Figure 7—figure supplement 2C

Figure 7J, Figure 8J, Figure 9S Figure 7—figure supplement 2D

Reviewer #3:1) Subsection “An updated inventory of proboscis muscles” (second paragraph). Either a semi-colon or full stop after m3M.

Done.

Also, in the figure legend, define L at the bottom.

Added to legend of Table 1.

2) In the Abstract, they write "…and throughout entire circuits, as the fly connectome approaches."

Now removed, per reviewer 1.

However, there is no mention of this theme in the Discussion section. Maybe a sentence or two somewhere in the Discussion section on how their identification of the muscles and the motor neurons might help decipher the meaning of the interneuronal connections that will be revealed in the connectome. I personally believe it will be enormously useful, because, as the authors write (Introduction), "…they are the outputs upon which higher order circuits must act."

Now a brief mention in the Discussion section.